# Deep learning reveals that multidimensional social status drives population variation in 11,875 US participant cohort

Justin Marotta[1,2], Shambhavi Aggarwal[1,2], Nicole Osayande[1,2], Karin Saltoun[1,2], Jakub Kopal[3], Avram J. Holmes[4], Sarah W. Yip[5,6], Danilo Bzdok[1,2,7,8]*

**1** McConnell Brain Imaging Centre, Montreal Neurological Institute (MNI), McGill University, Montreal, Quebec, Canada, **2** Mila - Quebec Artificial Intelligence Institute, Montreal, Quebec, Canada, **3** Centre for Precision Psychiatry, Division of Mental Health and Addiction, Institute of Clinical Medicine, University of Oslo, Oslo, Norway, **4** Department of Psychiatry, Brain Health Institute, Rutgers University, Piscataway, New Jersey, United States of America, **5** Department of Psychiatry, Yale University School of Medicine, New Haven, Connecticut, United States of America, **6** Child Study Center, Yale University School of Medicine, New Haven, Connecticut, United States of America, **7** Department of Biomedical Engineering, Faculty of Medicine, McGill University, Montreal, Quebec, Canada, **8** School of Computer Science, McGill University, Montreal, Quebec, Canada

* danilo.bzdok@mcgill.ca

## Abstract

As an increasing realization, many behavioral relationships are interwoven with inherent variations in human populations. Presently, there is no clarity in the biomedical community on which sources of population variation are most dominant. The recent advent of population-scale cohorts like the Adolescent Brain Cognitive Development[SM] Study (ABCD Study®) are now offering unprecedented depth and width of phenotype profiling that potentially explains interfamily differences. Here, we leveraged a deep learning framework (conditional variational autoencoder) on the totality of the ABCD Study® phenome (8,902 candidate phenotypes in 11,875 participants) to identify and characterize major sources of population stratification. 80% of the top 5 sources of explanatory stratifications were driven by distinct combinations of 202 available socioeconomic status (SES) measures; each in conjunction with a unique set of non-overlapping social and environmental factors. Several sources of variation across this cohort flagged geographies marked by material poverty interlocked with mental health and behavioral correlates. Deprivation emerged in another top stratification in relation to urbanicity and its ties to immigrant and racial and ethnic minoritized groups. Conversely, two other major sources of population variation were both driven by indicators of privilege: one highlighted measures of access to educational opportunity and income tied to healthy home environments and good behavior, the other profiled individuals of European ancestry leading advantaged lifestyles in desirable neighborhoods in terms of location and air quality. Overall, the disclosed social stratifications underscore the importance of treating SES as a multidimensional construct and recognizing its ties into social determinants of health.

**Data availability statement:** The data supporting the findings of this study are available from the Adolescent Brain Cognitive Development Study (ABCD Study®) dataset. The ABCD Study® is a publicly available resource accessible through the National Institute of Mental Health Data Archive (NDA). All relevant instructions to obtain the data can be found online (https://nda.nih.gov/abcd/request-access).

**Funding:** DB was supported by the Brain Canada Foundation, through the Canada Brain Research Fund, with the 1371 financial support of Health Canada, the Canadian Institute of Health Research (CIHR 438531, 1373 CIHR 470425), the Healthy Brains Healthy Lives initiative (Canada First Research Excellence fund), the IVADO R3 1374 AI initiative (Canada First Research Excellence fund), and by the CIFAR Artificial Intelligence 1375 Chairs program (Canada Institute for Advanced Research).

**Competing interests:** DB is a shareholder and advisory board member at MindState Design Labs, USA. This does not alter our adherence to PLOS ONE policies on sharing data and materials.

## Introduction

In biomedical research in humans, individual differences in demographic and other variables (e.g., effects of sex on brain volume) may often be stronger than variation in the effects of primary scientific interest (e.g., relationship between depression and brain volume) [1,2]. The integrity of analysis and interpretation may thus be jeopardized by insufficient acknowledgement of basic sources of variation that naturally occur in human populations (hereafter referred to as 'background variation'). Strong background variations previously identified include factors such as socioeconomic status (SES) [3], ancestry [4,5], height [6,7], and nutrition habits [8]. In the past, biomedical investigators attempted to avoid the issue of background variation influencing study findings by employing strict inclusion criteria [9,10] and by over-selection of Western, educated, industrialized, rich and democratic societies (WEIRD) [11]. However, these groups typically make up less than 15 per cent of the global population. This restrictive approach led to a homogenized, some would say artificially "cleaned", under-representative stratum of society that made up many study cohorts [4,12]; making it challenging to generalize findings and conclusions to the broader population. Over 10% of the US population is comprised of individuals from multiracial groups [13]. This percentage is on the rise, underscoring the imperative to revise modeling approaches which are able to embrace and explicitly model such increasing sources of heterogeneity.

The recent advent of big data in biomedicine is now opening the door to studies that include larger, more heterogeneous cohorts of individuals more reflective of the general population. However, appropriately dealing with the heterogeneity inherent in these participant cohorts is an unsolved challenge of everyday data analysis in biomedicine [14–17]. Modeling efforts aiming to capture nuanced brain-phenotype relationships from heterogeneous datasets consistently fail to do so [18]. For example, a study aiming to separate typically developing individuals from those with a brain disorder based on brain imaging features found decreased classification accuracy with increasing participant diversity [19]. Diversity due to major sources of population variation was captured using propensity scores as composite confound indices. Commonly used and new deconfounding techniques could not sufficiently mitigate the impact of increased diversity in participants samples. This provided evidence that many of today's default deconfounding techniques may be insufficient to deal with the effects of increasing heterogeneity in our study samples.

The challenge lies in our current lack of a comprehensive understanding of the primary factors influencing population-level differences and of a clear methodology for their identification. Historical limitations, such as the aforementioned lack of population representativeness in study cohorts, coupled with often small, underpowered samples and a narrow choice of variables, hamper our capacity to unveil influential factors; let alone rank them according to their overall importance.

The Adolescent Brain Cognitive Development[SM] Study (ABCD Study®) stands out as a unique data resource poised to overcome many shortcomings in understanding population-level differences. The ABCD Study® aims to track the brain development

and health of over 11,000 youth spanning 21 cities across the United States from ages 9–10 into young adulthood. As the largest study of its kind, the ABCD Study® cohort is intentionally designed to represent the broader US population [20] (cf. Methods regarding sampling strategy). Through expansive scale and nuance of phenotyping, the study enables detailed annotation of each participant's background. Here, we combine advanced statistical analyses – in the form of deep auto-encoder neural networks – with ABCD's unique data, seizing an unprecedented opportunity for these models to unearth major sources of interindividual differences, stratifying the population in ways that have not been previously observed or even measured; let alone aggregated into a holistic analysis.

Prior studies aiming to identify sources of interindividual differences key to dividing, or in other words, stratifying the population into distinct subgroups have not taken an appropriately multivariate or many-variable approach. A study applying Principal Component Analysis (PCA) [21] to US census data found a component of population variation indexing "socioeconomic disadvantage", driven by factors such as impoverished living conditions, low high school graduation rate, minoritized status, and single parent households [22]. The study considered information from approximately 312 million people based on census estimates, but only included fifteen variables selected to capture variations of social and environmental characteristics. While the first component was characterized by classic socioeconomic measures, subsequent principal components reflected additional dimensions of socioeconomic disadvantage related to limited mobility and disability, urbanicity and income, and immigrant groups. In light of these findings, the authors recommended adopting multidimensional measures that can capture the inherent complexity of factors influencing socioeconomic stature, as opposed to relying on single-dimensional measures.

Collectively, early studies can be viewed as touching on factors of population diversity related to social determinants of health (SDOH). As defined by the World Health Organization (WHO), SDOH encompass "the conditions in which people are born, grow, live, work, and age" [23]. Evolving evidence indicates that efforts to improve health outcomes targeting SDOH may yield more substantial impacts than interventions to improve health outcomes targeting individual biological and behavioral factors [24,25]. For example, it has been estimated that medical care accounts for only 10–20 percent of the modifiable contributors to healthy outcomes for a population, with the remaining 80–90 percent attributed to SDOH [26]. Moreover, the magnitude of impact is highlighted by estimates revealing that the number of deaths in the United States attributable to six SDOH rivals the total numbers attributed to leading pathophysiological causes such as heart attacks, strokes, and lung cancer [27].

Findings from prior research demonstrate that across US states and cities with equally low SES, communities with greater income inequality were linked with poorer health outcomes [28]. Building on this, a meta-analysis involving approximately 60 million participants demonstrated that individuals residing in regions with high income inequality faced an increased mortality risk independent of their SES, age, and sex [29]. Furthermore, several studies have found subjective social status to be at least as predictive of various health outcomes as objective measures of SES [30–32]. In other words, the perception of disadvantage may be an important reason why lower SES is linked with worse health outcomes. Extending to a cross species perspective, research indicates that robust links between social status and health also exist in non-human primates [33]. Such effects have been demonstrated [34] to stem from elevated physical and psychosocial stressors and lead to a breadth of adverse physiological effects [33]. These examples contribute to a growing body of evidence [28,35,36] suggesting psychosocial factors, a subset of SDOH which involve the interplay between social elements and individual cognition and behavior [37], can play an important role in the relationship between SES and health.

In the present study, by designing a data-driven machine learning framework, we systematically identified key sources of population stratification that track interindividual differences. We took a bottom-up approach, leveraging the full range of the unusually rich phenotyping within the ABCD Study® population dataset. In contrast to the common approach of hand-picking variables to look for effects based on a priori research intuitions derived from selected participants, our approach is comprehensive. That is, we analyzed the entire range of phenotypes from the ABCD Study® initiative, aiming to determine which phenotypic combinations conjointly emerge as the most prominent sources of variation across 21

US cities sampled in this cohort of over 11,000 participants. Our deep learning architecture provides an advantage over commonly used dimensionality reduction techniques in that it has the potential to uncover complex non-linear patterns of covariation between data variables – a characteristic known to exist in human population modeling [38,39]. Utilizing a deep learning architecture to disentangle the general structure of population-scale phenotypical variation in a massively phenotyped US cohort marks an uncharted frontier and a novel approach within our study. Our holistic approach has the potential to enhance future research endeavors by directing studies toward pertinent measures to include when investigating a phenotype of interest. Ultimately, our findings take a step towards fostering diversity-aware modeling in biomedical research.

## Results

Initially, as a preparatory step before identifying major sources of interfamily differences, we compared several candidate deep learning model configurations. We applied a preprocessing pipeline (cf. Methods) to the entirety of the ABCD Study® 4.0 release phenotypic data, leading to 8,902 curated variables for 11,875 participants as the starting point for our downstream analyses (Fig 1; Phase 1). These cleaned data were split into training (80% of the participant sample), validation (10% of the sample), and test (10% of the sample) datasets and used to train our Conditional Variational Autoencoder (CVAE) architecture. The CVAE was selected as our deep learning architecture for its ability to learn complex relationships between input variables while explicitly acknowledging site-related variation by conditioning on the participant's ABCD Study® data collection site.

To identify the best model architecture, we constructed a disciplined model comparison pipeline in which several model configurations were systematically benchmarked across randomized model weight initializations (Fig 1; Phase 2.a). We selected the best model architecture based on lowest achieved mean squared error (MSE) reconstruction performance on the independent participants (untouched 10% of the sample, test split). Performance of our best model architecture was validated by comparing MSE reconstruction loss on these unseen participants against that of Principal Component Analysis (PCA) – the most commonly used dimensionality reduction technique, suggesting itself as a baseline model to compare against (Fig 1; Phase 2.b). Importantly, the CVAE mean reconstruction performance (mean MSE = 865.9, SD = 1.09 across different model initializations) exceeded the reconstruction performance of PCA (mean MSE = 871.9, SD = 0.20) by more than 2 standard deviations with both methods using equal latent space dimensionality.

To accurately reconstruct the data, the patterns and structure learned by the latent variables of our model, hereafter referred to as 'components', aimed to capture the most important sources of variation across participants in the data. The better reconstruction performance exhibited by our approach implied that our deep learning framework has extracted the underlying patterns and structure of the data with more fidelity than PCA. Therefore, because our model reconstructed the input data more accurately than PCA, by a more faithful internal representation, we can interpret its learned components as sources of variation key to stratifying the US population.

To identify the top components among our 100-dimensional latent space (cf. Methods), we ranked the components of our optimal performing model by amount of variance explained (Fig 1; Phase 2.c). We then applied the widespread elbow criterion, which determined that the 10 leading components account for a higher proportion of variance in the data than the subsequent components that could be extracted and interpreted. To give a quantitative answer to the question 'How much is component j reflected in a given family's phenome?', we calculated the participant-wise expressions in each CVAE component of population variation by passing a participant's phenome fingerprint (an 8,902-item vector) through the encoder neural network and sampling from latent space. These participant-wise expressions are henceforth referred to as scores. To understand which phenotypes played prominent roles in each of the identified leading components, we computed phenotype loadings in each component by estimating Pearson's correlation coefficients between respective CVAE component scores and the original phenotype measurement across study participants. Henceforth, we refer to the absolute value of these phenotype loadings as 'weight strengths'.

**Phase 1: Data collection & preprocessing**

- Download ABCD release 4.0 behavioral data
- Data cleaning (missing values, outliers, continuous and discrete variables, z-score, one-hot encode)
- Final dataset size = (11875 subjects, 8902 phenotypes)

**Phase 2: Model tuning & validation**

a) Hyperparameter tuning, train with early stopping

$\mathcal{L}_{CVAE}(x, \hat{x}, \mu, \sigma)$

Conditions, $c$

b) Validate performance against principal component analysis (PCA)

c) Apply elbow criterion to identify most explanatory components

**Phase 3: Interpretation**

- Group phenotypes into 23 predefined categories
- Threshold each phenotype at 95th percentile weight strength across components
- Top 10 most explanatory components named A-J
- Plot by category mean weight strength, examine top phenotypes in driving categories

**Fig 1. Analytical protocol: Deep learning enables identification of key phenotype groups driving interindividual differences from richly profiled population scale dataset.** Our study can be broken into three broad phases. Phase 1: We included the entirety of the ABCD Study® release 4.0 phenotypic data in our analysis pipeline. Data were preprocessed to retain as rich a phenotype profile as possible across all available participants, while preserving true distributions and handling outliers (cf. Methods). The final dataset size post-preprocessing steps was 11,875 participants each with 8,902 phenotypical variables. Phase 2: **a)** We trained a Conditional Variational Autoencoder (CVAE) across a range of hyperparameters (cf. Methods)

and benchmarked their performance against Principal Component Analysis (PCA). **b)** The best performing CVAE architecture mean reconstruction loss was more than 2 standard deviations below that of PCA (CVAE mean MSE = 865.9, SD = 1.09; PCA mean MSE = 871.9, SD = 0.20). **c)** Components of the best performing CVAE architecture were ranked in terms of a heuristic per-component explained variance metric (cf. Methods) and the widespread elbow criterion was applied to determine that 10 components account for a high proportion of variance in the data compared to the remainder of the top 100 components. This indicated that key modes of population stratification exist, and we focused on these 10 most explanatory components for interpretation. Phase 3) We computed the 95th percentile among all 100 components to retain phenotypes only in the 5 components where they exhibit the highest weight strength, this enabled us to reduce the number of variables to only the most important for characterizing each component. Grouping the remaining phenotypes in each of the top 10 components (A-J) into 23 predefined categories provided by the ABCD Study®, we identified driving categories per component by calculating the mean weight strength per category in each component. Driving categories were examined in further detail to identify which particular phenotype groups were conjointly responsible for driving the captured population variation.

The most important phenotypes for differentiating each of our learned components were identified by thresholding to retain each phenotype only in components where it had a weight strength in the 95th percentile across all 100 components (cf. Methods). In other words, each phenotype is retained only in the 5 components where it exhibits the largest weight strength. For each of the top 10 ranked components (identified as A-J), we then grouped the retained phenotypes into 23 predefined categories (e.g., Socioeconomic, Behavior, Demographics, etc.) provided by the ABCD Study®. By ranking the phenotypes in each category by mean weight strength, we found the most representative categories in each of the 10 leading components (Fig 1; Phase 3).

As illustrated by the variation in category mean weight strengths across all the top components (Fig 2.a), our approach was able to capture distinct groupings of interlocking categories of phenotypes driving population variation. The Socioeconomic category (SES) was found to exhibit the strongest or second strongest mean weight strength of all categories in four of the top five most explanatory components, A, B, D, and E. Furthermore, these four components exhibited the largest mean SES phenotype weight strength of all top 10 components (Fig 2.b) and retained the highest proportion of SES phenotypes after thresholding for representativeness (Fig 2.c). These SES-centric components were the focus of our interpretation. Exact quantities obtained for weight strength and proportion of SES phenotypes across the top 10 components can be found in Table 1. The Neuropsychological Tests category also exhibited consistently large mean weight strength across all top components.

Component A was dominated by SES phenotypes in conjunction with phenotypes mainly from Mental Health Summary, Neuropsychological Tests, Social Responsiveness, and Demographics based on mean weight strength (mean Pearson's $|r|$ = 0.40, 0.22, 0.21, 0.17, and 0.15 respectively). Top categories in component B in terms of mean weight strength included Neuropsychological Tests, SES, Parent Characteristics, Social Adjustment, Diagnostic (KSADS), and Behavior (mean Pearson's $|r|$ = 0.31, 0.25, 0.16, 0.152, 0.149, and 0.14 respectively). In component D, the most salient categories included Neuropsychological Tests, SES, Diagnostic (KSADS), Nutrition, Demographics, and Cognitive Capacity (mean Pearson's $|r|$ = 0.28, 0.20, 0.14, 0.13, 0.13, and 0.13 respectively). For component E, the most prominent categories included SES, Neuropsychological Tests, Mental Health Summary, Demographics, and Social Adjustment (mean Pearson's $|r|$ = 0.34, 0.20, 0.177, 0.174, and 0.171 respectively). The complete list of phenotypes mean weight strength and proportion per category in each of the top 10 components can be found in S1 Table and S2 Table in S1 File, respectively.

Considering the emergence of SES as a key category that cuts across four of the top five components, we wanted to examine to what extent each of our discovered SES-centric components captured distinct dimensions of SES. Of the totality of 202 candidate SES measures, we examined the number of measures that were commonly represented (exceeding the 95th percentile thresholding) in each of the 4 SES-centric components. We found that no single measure among the 202 measures was present in all 4 components, and that each of the components presented a unique configuration of flagged SES measures not captured in any of the other 3 components (Fig 3). The complete tables of unique measures per component are available in the code repository https://github.com/dblabs-mcgill-mila/abcd_dl_analysis.

In component A, uniquely identified SES measures included living in low-income areas (Pearson's $|r|$ = 0.43) with elevated levels of unemployment (Pearson's $|r|$ = 0.44) and people living below the poverty threshold (Pearson's $|r|$ = 0.56).

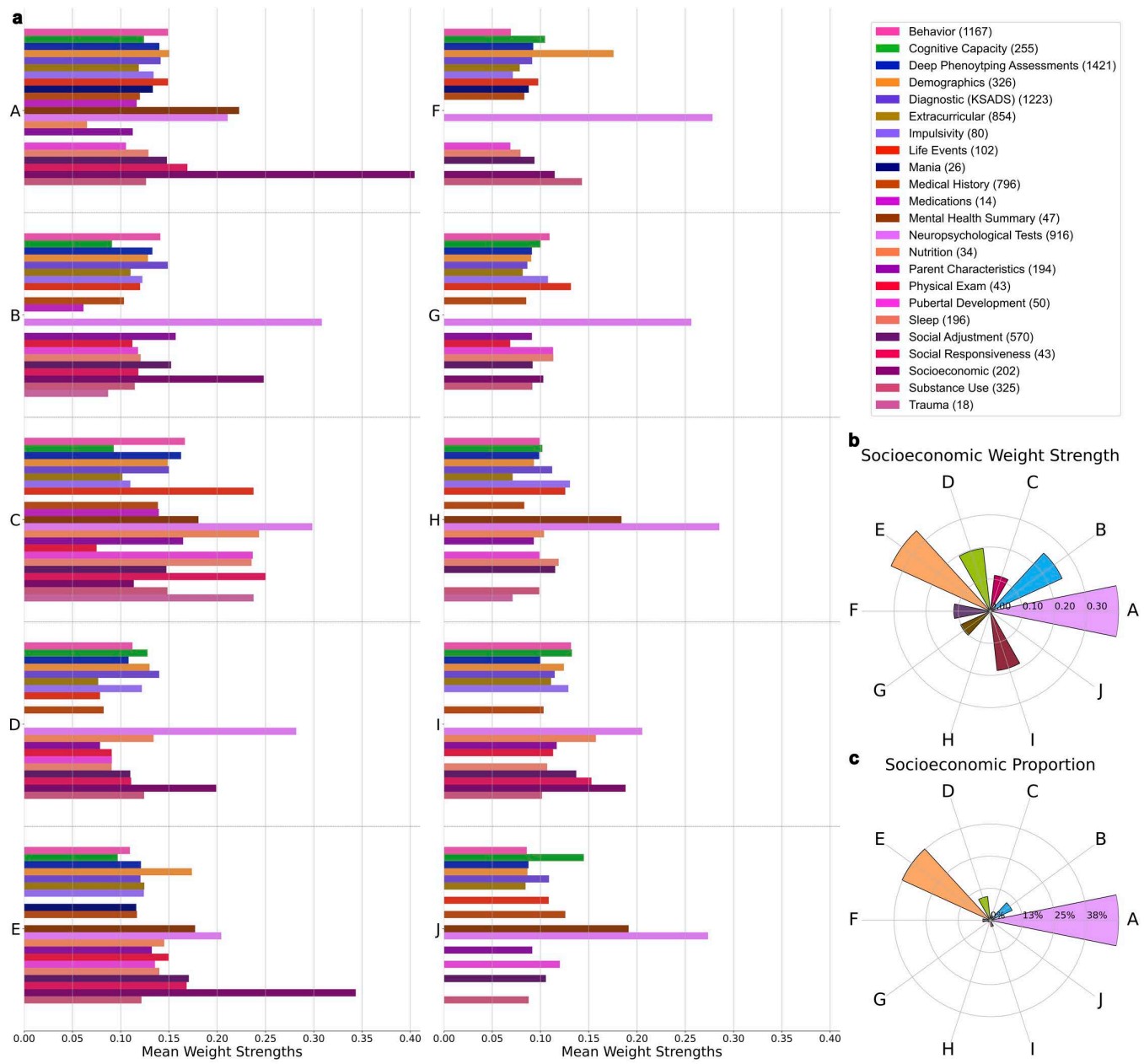

**Fig 2. Population variation is driven by distinct combinations of phenotype categories with SES as a central theme. a)** Weight strength colored by category in each of the top 10 components. The number of variables per category is listed in brackets next to each category name in the legend. Category mean weight strength is calculated by averaging all individual phenotype weight strengths (retained after thresholding) within a predefined category. The Socioeconomic (SES) category has the highest or second highest mean weight strength of all categories in 4 of the top 5 most explanatory components. The Neuropsychological Tests category also exhibits consistently high mean weight strength across the top 10 components. **b)** Radial plot illustrating that the SES category has the highest mean weight strength in 4 of the top 5 most explanatory components (A, B, D, E) compared to the remainder of the top 10 components. **c)** Radial plot revealing that the phenotypes driving the SES category are distributed amongst 4 of the top 5 most explanatory components (A, B, D, E) in a higher proportion compared to the remainder of the top 10 components.

**Table 1. Mean weight strength and proportion of phenotypes from the Socioeconomic (SES) category across the top 10 ranked components. Components A, B, D, and E are bolded because they exhibit both the highest mean weight strength and proportion of SES phenotypes.**

| Component | SES phenotype weight strength [mean Pearson's \|r\|] | SES phenotype proportion [%] |
|---|---|---|
| **A** | **0.404** | **50.5** |
| **B** | **0.248** | **9.4** |
| C | 0.114 | 2.0 |
| **D** | **0.199** | **9.4** |
| **E** | **0.343** | **38.1** |
| F | 0.115 | 3.0 |
| G | 0.103 | 1.0 |
| H | 0 | 0 |
| I | 0.188 | 2.5 |
| J | 0 | 0 |

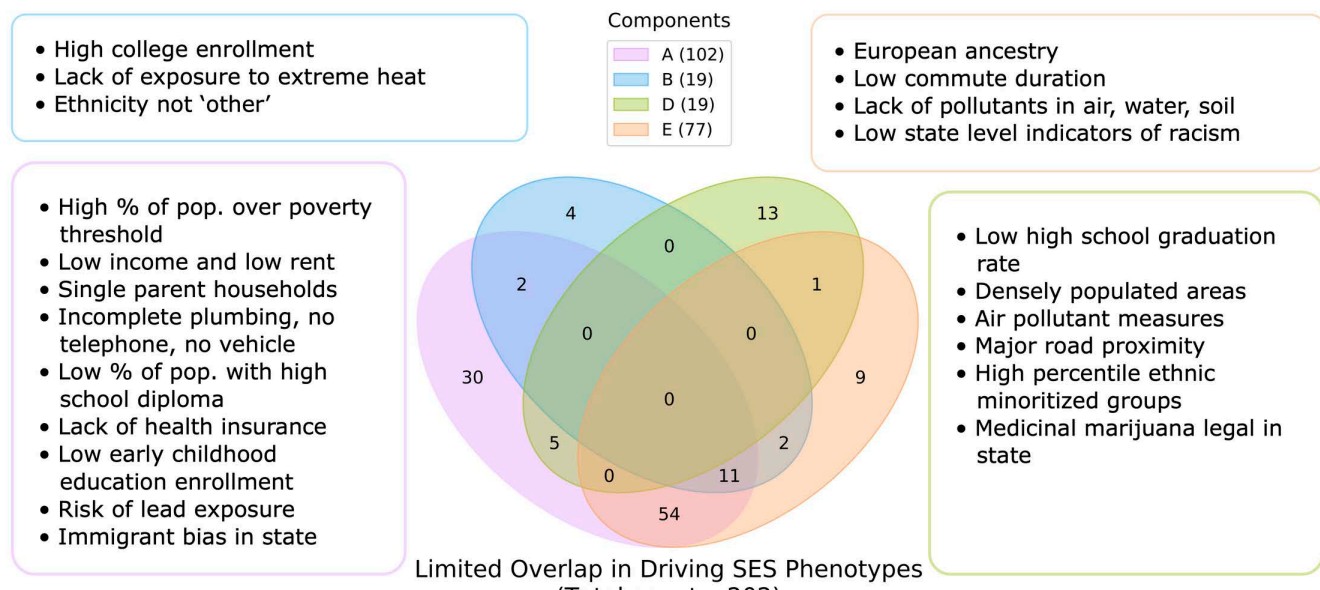

**Fig 3. Socioeconomic status is characterized by distinct constellations of phenotypes.** Each of the 4 Socioeconomic (SES) driven components presents unique measures not captured in any of the other 3 components. Out of 202 candidate SES measures, there are no phenotypes shared among all 4 of the components after thresholding to retain phenotypes in components where they rank in the 95th percentile (number of SES phenotypes retained in each component is listed in brackets next to each component name in the legend). Unique SES measures in component A relate to material poverty and health risks. SES measures solely captured in Component B relate to educational level and temperate climate. Component D uniquely captures measures related to densely populated living and areas with a high percentile of racial and ethnic minoritized groups. Component E is uniquely driven by European ancestry and measures of healthy and desirable environments.

Other distinct measures in component A tracked low scores on health and environment indices such as neighborhoods with a high risk of lead exposure (Pearson's $|r| = 0.33$) and high percentage of the population without health insurance (Pearson's $|r| = 0.37$). Additional unique neighborhood measures included houses with incomplete plumbing (Pearson's $|r| = 0.17$), no telephone (Pearson's $|r| = 0.28$), and no vehicle (Pearson's $|r| = 0.39$). From a social standpoint, these families

lived in areas with a high percentile of single parent households (Pearson's |r|=0.43). In contrast, SES measures solely captured in Component B related to educational level and temperate climate via metrics tracking living in areas with college enrollment in nearby institutions (Pearson's |r|=0.14), and lack of extreme heat exposure (Pearson's |r|=0.13). Component D uniquely captured markers of high gross residential and population density (Pearson's |r|>0.21) and areas with a high percentile of racial and ethnic minoritized groups (Pearson's |r|=0.28). Specific racial and ethnic groups considered as part of this measure are outlined in the CDC SVI Documentation [40]. These are accompanied by educational measures such as areas with low high school graduation rate (Pearson's |r|=0.15) and low percentage of 9th graders graduating high school on time (Pearson's |r|=0.15). Unique measures in Component E included proportion of European ancestry (Pearson's |r|=0.40) and measures of healthy and desirable environments such as low commute duration (Pearson's |r|=0.13), as well as lack of pollutants in air, water, and soil (Pearson's |r|=0.12).

Having identified that each of the four SES-centric components exhibited a distinct profile of SES phenotypes, we next examined sets of co-occurring driving phenotypes within each of these components of population variation across categories. Rather than focusing on distinct SES measures only, we considered phenotypes with the highest weight strength across all categories within a given component. The aim here was to identify the groups of phenotypes driving each component in conjunction with the characteristic SES measures, to understand if a component characterized by distinct SES measures would also be distinguished by unique co-occurring sets of phenotypes from other categories. We found that each SES-centric component exhibited a distinct set of conjointly relevant driving phenotypes. In this section, we outline findings for these four SES-centric components, weight strengths of driving phenotypes in all 100 components are available in the code repository https://github.com/dblabs-mcgill-mila/abcd_dl_analysis.

Our findings in component A (Fig 4) propose a connection between economic resources, social and behavioral issues, and mental and emotional well-being. Collectively, considering top categories, top phenotypes, and unique SES phenotypes, component A distinctly captured material poverty and its health and mental well-being correlates. The strongest SES phenotypes in component A related to neighborhoods low in economic resources (Pearson's |r|=0.59), child opportunity level (Pearson's |r|=0.57), and high in poverty rate (Pearson's |r|=0.56) and public assistance rate (Pearson's |r|=0.56). Cognition related phenotypes exhibiting strong weight in this component related to the child performing poorly on working memory (Pearson's |r|=0.38) and reasoning tasks (Pearson's |r|=0.30). Phenotypes from the Mania category were well represented in component A, in terms of highest mean weight strength (Pearson's |r|=0.13) and in the highest proportion (53.8%) of all components. Measures included youth reported experiences of extreme happiness, intense energy, and rapid thoughts (Pearson's |r|>0.13). Other emotional wellbeing measures tracked by component A included parents assessing their child as exhibiting swings in feelings and energy (Pearson's |r|=0.28) as well as indicators of mania and depression (Pearson's |r|=0.38). These were accompanied by parent-reported restlessness indicators such as difficulty waiting turn, acts like a driven motor, and fidgeting (Pearson's |r|>0.23), as well as parent reported youth issues with social interaction and communication difficulties as measured by avoiding eye contact, maintaining rigid routines, concentrates too much on parts of things rather than seeing the whole picture, difficulty understanding tone of voice and facial expressions (Pearson's |r|>0.19). Deviant social behavioral problems were reported by parents through the Child Behavioral Checklist (CBCL) scales relating to social problems, rule breaking, and stress (Pearson's |r|>0.30). Other phenotypes emphasized in component A were Black ethnicity (Pearson's |r|=0.30), neighborhood risk (Pearson's |r|=0.24), and high youth reported screen time (Pearson's |r|=0.27).

The overall profile of component B (Fig 5) highlighted the breadth of social and environmental factors, like local neighborhood and family context, impacting educational attainment and mental performance outcomes such as working memory. The theme of this component can be summarized as relating educational and behavioral outcomes shaped by one's upbringing. In component B, the most dominant SES phenotypes included neighborhood measures relating to child educational opportunity level (Pearson's |r|=0.37), low rates of disability status (Pearson's |r|=0.31), third grade reading proficiency (Pearson's |r|=0.30), and being in the 1st percentile of national household income ranking (Pearson's

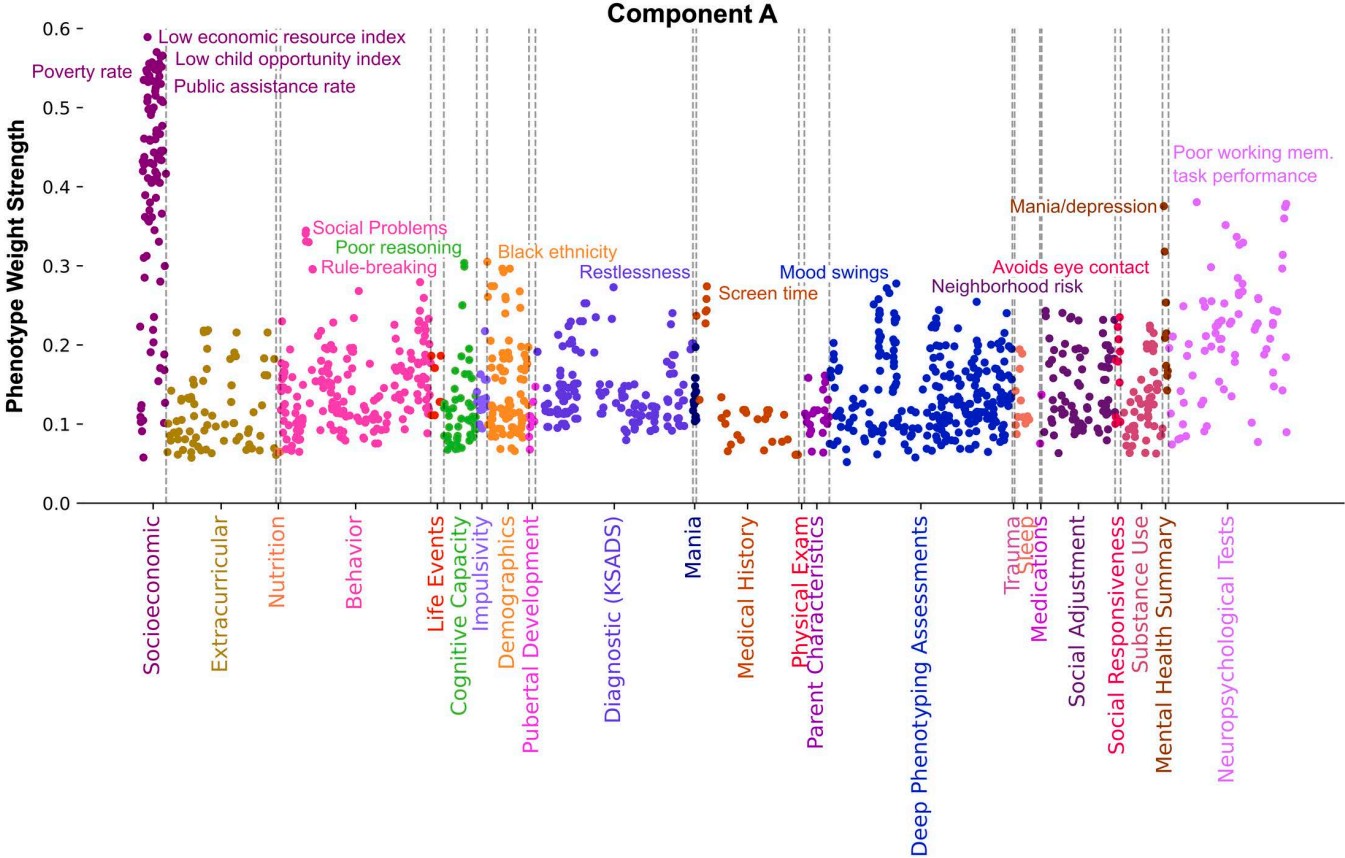

**Fig 4. Component A distinctly captures material poverty and its health and mental well-being correlates.** Manhattan plot shows phenotypes, colored by category, whose weight strength in component A rank in the 95th percentile among all 100 components. Weight strength is calculated as the magnitude of Pearson's correlation coefficient between participant data variable and latent variable scores. Socioeconomic measures exhibiting the strongest weight relate to high poverty and public assistance rates as well as low economic resources and child opportunity levels. Other phenotypes exhibiting strong weight relate to poor working memory task performance, poor reasoning, mood swings, indicators of mania and depression, restlessness, social interaction difficulties, rule breaking behavior, Black ethnicity, neighborhood risk, and high screen time. Collectively, this component proposes a connection between economic resources, deviant social behavior, and mental and emotional well-being.

|r| = 0.28). Good working memory task performance (Pearson's |r| = 0.57) was tracked by this component. Drivers in the Parent Characteristics category suggest nonviolent home environments with measures from parent surveys such as family members do not hit each other (Pearson's |r| = 0.23), family members do not one-up or outdo each other (Pearson's |r| = 0.19), and family members do not get so angry they throw things (Pearson's |r| = 0.15). Parents in this component also rated their children as possessing good concentration skills (Pearson's |r| = 0.26) and exhibiting good behavior as measured by playing quietly, not lying often, not participating in vandalism or bullying, and not destroying things belonging to their family or others (Pearson's |r| > 0.21). Other family context metrics in component B related to the family being non-religious (Pearson's |r| = 0.27) as measured by negative responses to God being first and family second, and that children should be taught to pray.

Strongly weighted phenotypes characterizing component D (Fig 6) highlight a connection between homeownership, living conditions, urbanicity, education, language, ethnicity, and recent immigrant status. These, together with the unique SES measures and dominant categories, can be summarized as illuminating the immigrant and racial and ethnic minoritized group experience in the USA. SES drivers indexed environments with low homeownership rates (Pearson's

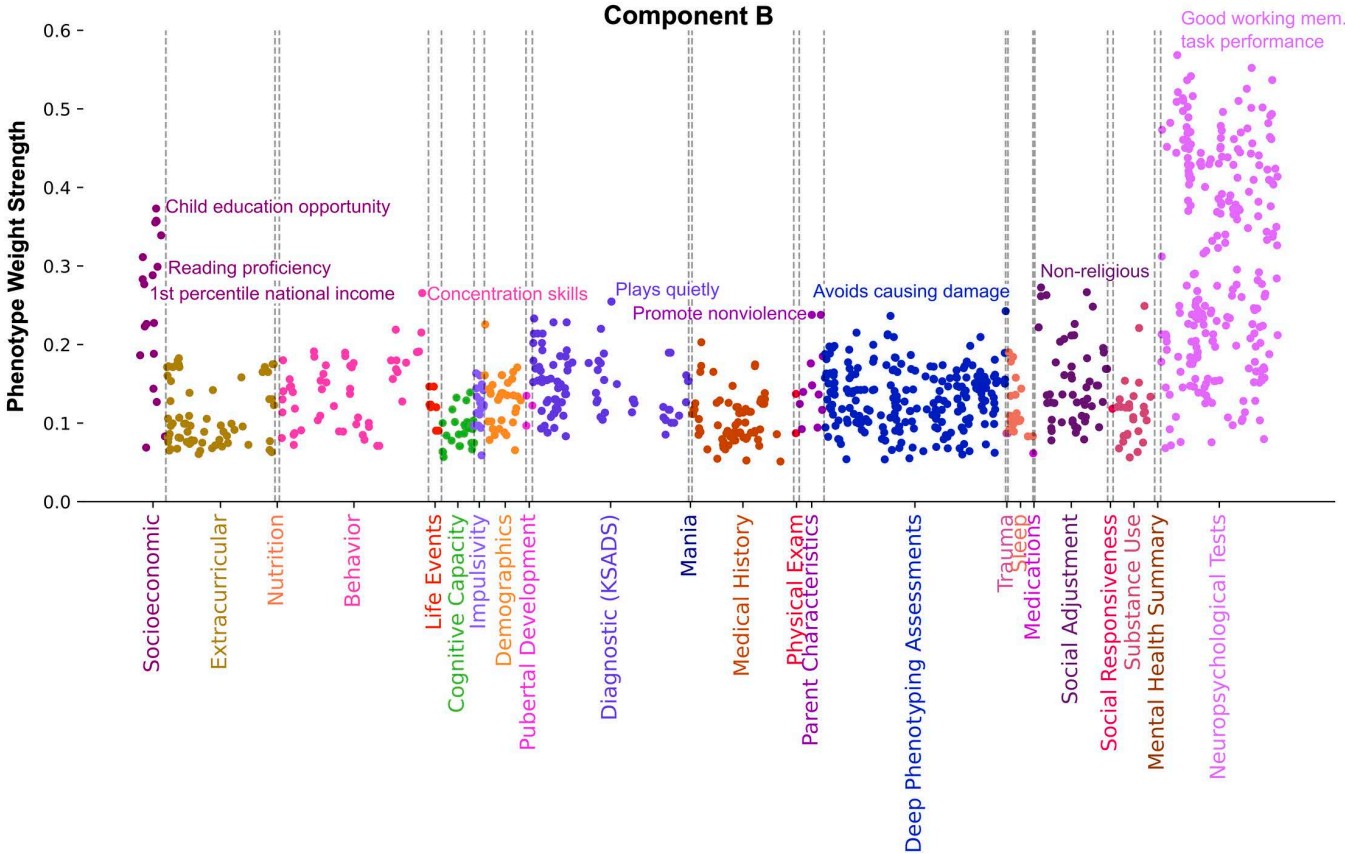

**Fig 5. Component B relates educational and behavioral outcomes shaped by one's upbringing.** Manhattan plot shows phenotypes, colored by category, whose weight strength in component A rank in the 95th percentile among all 100 components. Weight strength is calculated as the magnitude of Pearson's correlation coefficient between participant data variable and latent variable scores. Strongest weight Socioeconomic phenotypes in this component relate to a child's educational opportunity and family income. Strongly weighted phenotypes from other categories include good working memory task performance, being non-religious, possessing good concentration skills, having parents that promote nonviolence, and exhibiting good behavior as measured by playing quietly and avoiding causing damage. In summary, this component underscores the wide range of social and environmental influences that affect both educational achievement and behavioral outcomes.

|r| = 0.34), high percentile racial and ethnic minoritized groups (Pearson's |r| = 0.28), and urbanicity as measured by high population density and housing density (Pearson's |r| > 0.21). Component D emphasized phenotypes related to poor working memory task performance (Pearson's |r| = 0.53) and poor reasoning (Pearson's |r| = 0.30). The Demographics category included families who reported recently immigrating to the US (Pearson's |r| > 0.22) and whose native tongue was Spanish (Pearson's |r| > 0.22). Measures of family identifying their race as "Other", "Don't Know" or "American Indian" (Pearson's |r| = 0.11, 0.1, 0.08 respectively) exhibited weight strength in the 95th percentile in component D among all 100 components. These measures were accompanied by the parent reporting their child having a phobia (phobic object) (Pearson's |r| > 0.20) and exhibiting fearful and anxious behavior (Pearson's |r| > 0.22). Though mean weight strength was large, the Nutrition category captured a single measure regarding the parent reporting their child does not eat less than 1 tablespoon of butter a day (Pearson's |r| > 0.13).

The phenotypes driving component E (Fig 7) suggested a link between opportunity, lifestyle and belonging to elevated sociodemographic groups. Considering the distinct SES measures, dominant categories and individual phenotypes of component E, this component can be summarized as capturing the values and opportunities of those leading wealthy

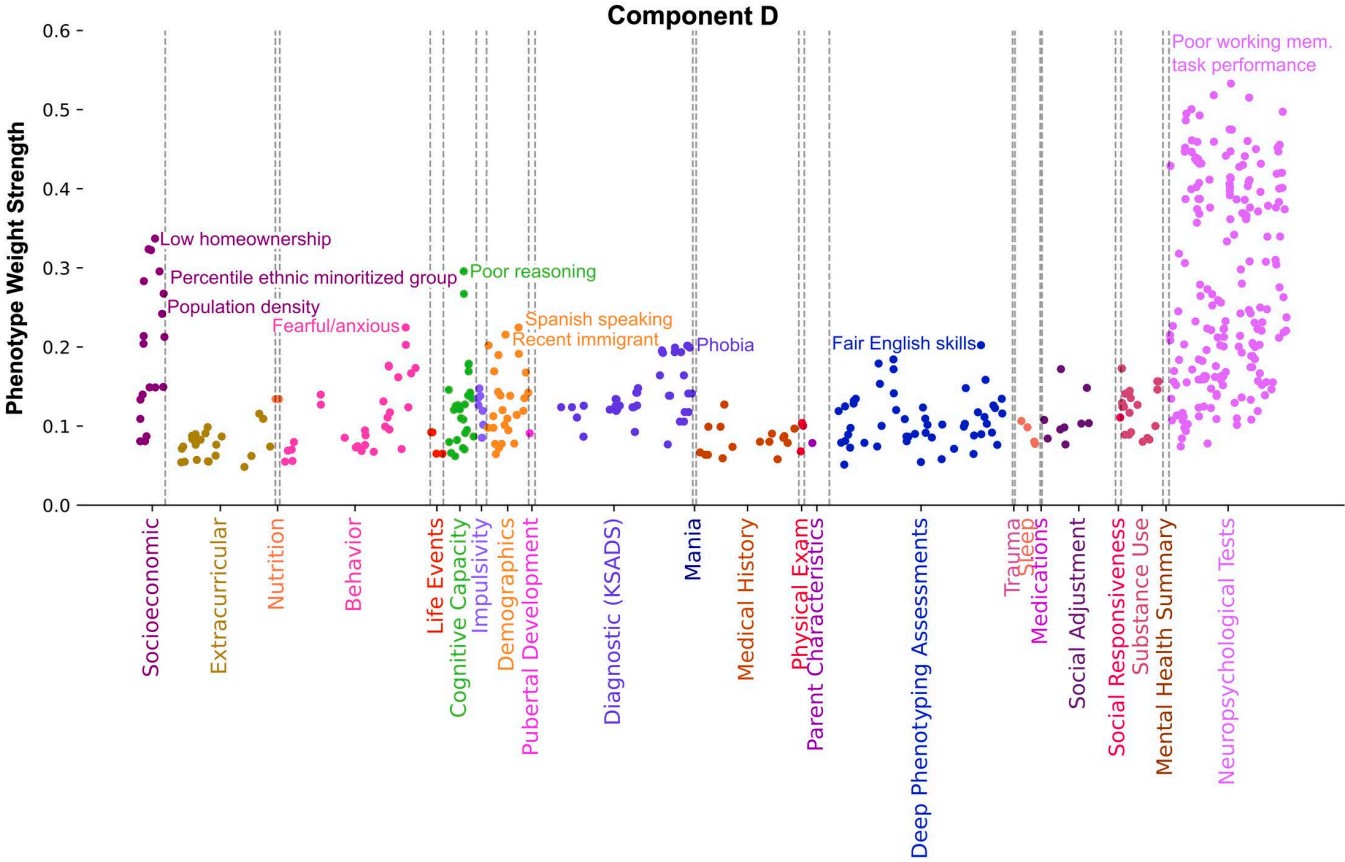

**Fig 6. Component D illuminates the experience of individuals as immigrants or members of racial and ethnic minoritized groups in the USA.**
Manhattan plot shows phenotypes, colored by category, whose weight strength in component A rank in the 95th percentile among all 100 components. Weight strength is calculated as the magnitude of Pearson's correlation coefficient between participant data variable and latent variable scores. Driving Socioeconomic and Demographics phenotypes include low homeownership rates, high population density and housing density, recently immigrating to the US, Spanish as native tongue, and belonging to a racial and ethnic minoritized group. These factors are accompanied by elevated levels of fear, anxiety, phobia, poor working memory task performance and reasoning. Together, these factors highlight a connection between living conditions, language, ethnicity, and immigrant status.

lifestyles. The most strongly weighted SES phenotypes in component E tracked neighborhoods high in terms of overall child opportunity (Pearson's $|r| = 0.45$), school wealth (Pearson's $|r| = 0.43$), low percentage of students eligible for free lunches (Pearson's $|r| = 0.41$), and proportion of European ancestry (Pearson's $|r| = 0.40$). Driving phenotypes in the Neuropsychological Tests category related to good working memory task performance (Pearson's $|r| = 0.36$) and good performance on the Little Man Task of visuospatial processing and attention (Pearson's $|r| = 0.33$). The Social Adjustment category driving phenotypes related to families with a lack of adherence to Mexican American cultural values such as religiosity, family obligation and family centricity (Pearson's $|r| = 0.40$). Component E had the highest mean weight strength (Pearson's $|r| = 0.12$) and retained the highest proportion of phenotypes from the Extracurricular category (24.7%) of all components. Extracurricular measures related to playing sports such as soccer, swimming, skiing or snowboarding, and baseball (Pearson's $|r| > 0.20$) or musical instruments such as guitar, piano, drums, violin, flute, and choir (Pearson's $|r| = 0.26$). Other strongly weighted phenotypes included student gets A's in school (Pearson's $|r| = 0.24$), parent financial responsibility as measured by paying debts and not having trouble managing money (Pearson's $|r| > 0.22$), White ethnicity (Pearson's $|r| = 0.36$), married parents (Pearson's $|r| = 0.31$), and low screen time (Pearson's $|r| = 0.24$). Though mean

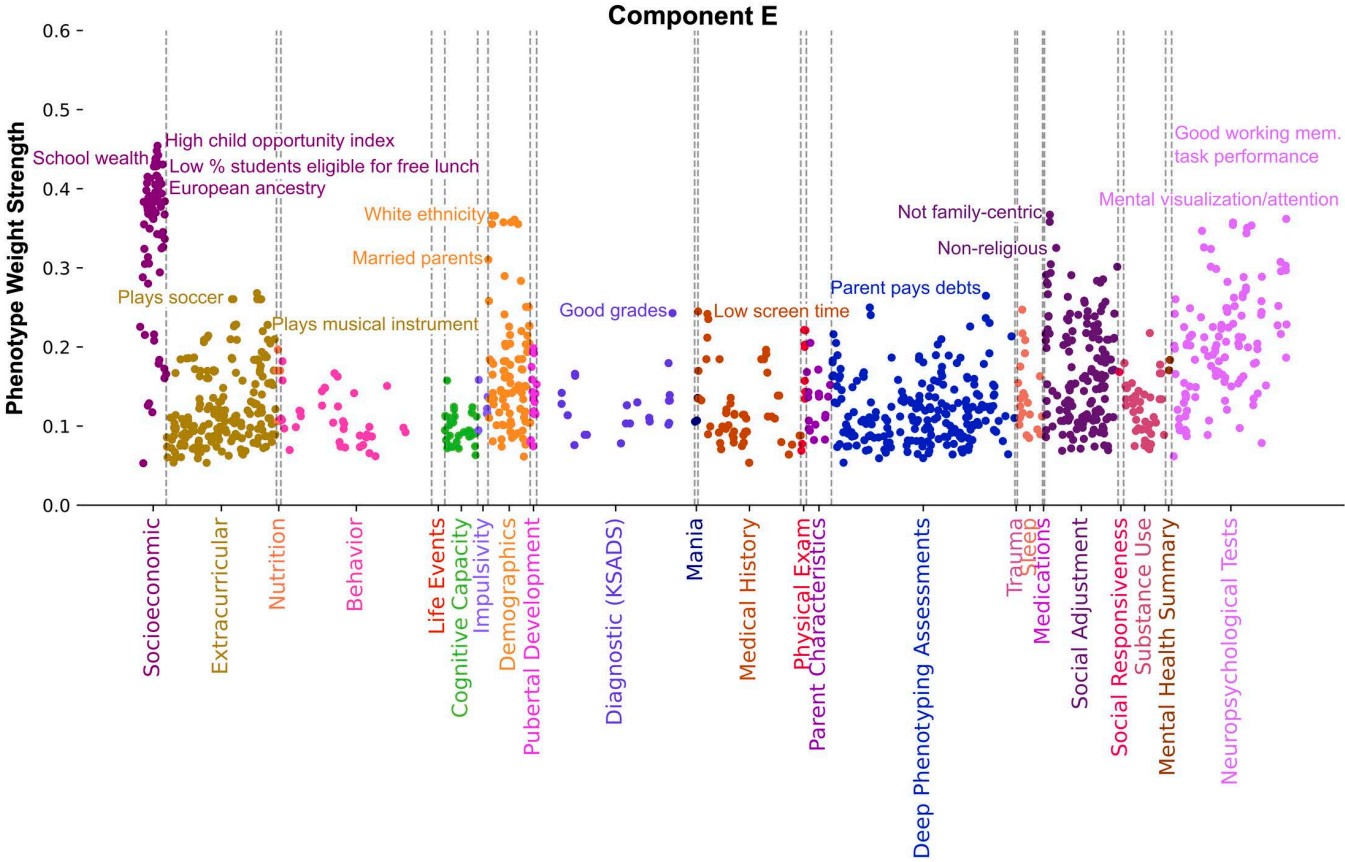

**Fig 7. Component E captures the values and opportunities of those leading wealthy lifestyles.** Manhattan plot shows phenotypes, colored by category, whose weight strength in component A rank in the 95th percentile among all 100 components. Weight strength is calculated as the magnitude of Pearson's correlation coefficient between participant data variable and latent variable scores. Driving Socioeconomic phenotypes include proportion of European ancestry, neighborhoods high in terms of overall child opportunity, school wealth, and a low percentage of students eligible for free lunches. Other strongly weighted phenotypes relate to good performance on tasks measuring working memory and visuospatial processing and attention, lack of Mexican American cultural values, good grades in school, parent financial responsibility, White ethnicity, married parents, low screen time, and involvement in extracurricular activities such as playing sports and musical instruments. This component illuminates a link between opportunity, lifestyle, and sociodemographic group.

weight strength was large, the Mental Health Summary category only contained two measures in component E: youth scoring low on positive urgency (Pearson's $|r| = 0.18$) and mania indicators (Pearson's $|r| = 0.17$).

Having established that components characterized by distinct SES measures can also be distinguished by their unique co-occurring sets of phenotypes, we wanted to provide an example illustrating the importance of treating socioeconomic status in a multidimensional manner as identified by our approach. To do so, we trained a multi-class logistic regression classifier to predict participant state residency solely using participant SES variable scores (n = 202) in components A, B, D and E as the input features available to the model. This signal alone was enough to distinguish ABCD Study® participants' state of residence with above chance level accuracy for 10 of 17 states (17 classes therefore chance level is ~5.88%). Logistic regression coefficient magnitudes per feature (participant SES scores in each component) were used to measure component predictive strength for each state. From this we determined that the most predictive SES component varied across states and that the degree of divergence between participant component score distributions differed across states (Fig 8). Successfully discerning participants' state of residency solely through the component scores of SES variables is

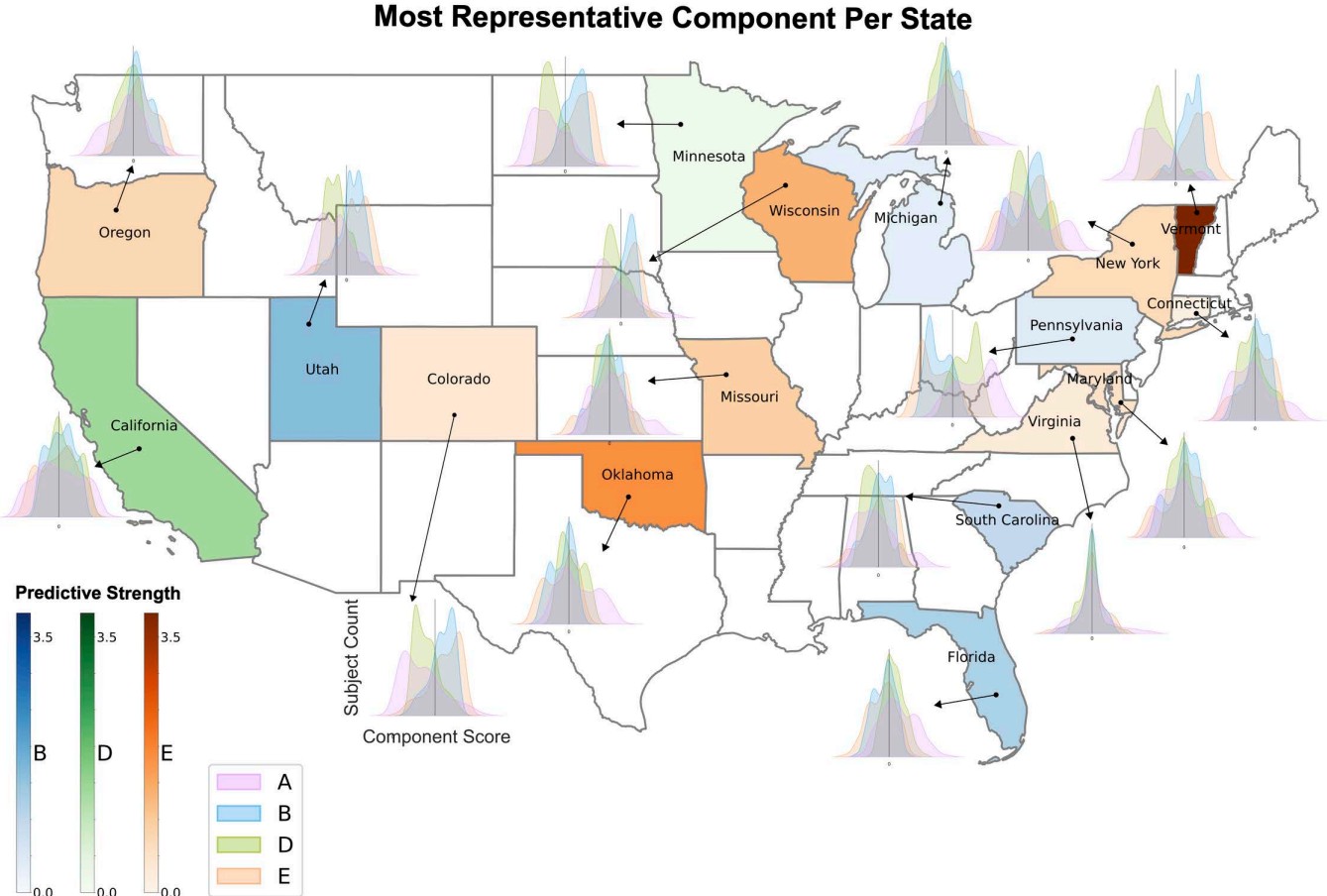

**Fig 8. Unique SES signatures are distinctly represented in specific US states.** Participant SES variable scores (n = 202) in each of the 4 SES-centric components were used as features to train a multi-class logistic regression classifier to predict participant state of residence (cf. Methods). State color indicates which SES component had the strongest influence (i.e., coefficient magnitude) on predicting state of residence and shading intensity of state quantifies the magnitude of this influence. Note that component A was not most influential for any state. Different states appear to be more uniquely aligned with different SES signatures. Density plots per state show the distribution of participant scores in each of the 4 components for that state. The degree of divergence between participant score distributions varies depending on the state. The classifier was able to predict US state of residence for 10 of 17 states at above chance level (17 classes therefore chance level is ~5.88%). Map using U.S. Census Bureau Cartographic Boundary File [41].

not only a validation of the distinctive dimensions of SES captured by our approach but also underscores the significance of viewing SES as a multi-dimensional construct.

## Discussion

### Unveiling the intricacies of interindividual differences through SES

Our present study was motivated by the lack of clarity on what dimensions of interindividual differences account for the most prominent variation across the expansive human phenome. It has been, and still is, common practice to carry out boutique studies with strict inclusion criteria and modest participant diversity. Consequently, neuroscience communities face challenges establishing analytical frameworks and sufficiently rich datasets to even begin to identify major sources of population stratification that drive interindividual variation. Now, we have datasets of unprecedented scale and phenotypic depth that provide fine grained accounts of each participant's multifaceted background and everyday life's circumstances.

The ABCD Study® exemplifies this kind of neuroscience dataset, an initiative aimed to be representative of the broader US population [20], encompassing ~11,000 participants and their family systems meticulously profiled by 8,902 distinct variables. As the most important observation across analyses, measures within the socioeconomic status (SES) domain emerged as key drivers in 4 of the top 5 leading components of population variation.

Here, we tailored a recently emerged deep representation learning framework, the conditional variational autoencoder, to the unique ABCD Study® resource to begin to identify yet-to-be-discovered major sources of interindividual differences in a disciplined approach. The ABCD Study® stands out as one of the most comprehensive cohort studies available today, tracking over 11,000 participants and their family systems across 21 US cities. The study captures an extensive array of phenotypes through a multitude of trusted batteries and gold standard assessment instruments. This wealth of diverse data combined with our deep learning framework empowered us to see deeper into various factors that underlie population variation, shedding light on intricate constellations, correlations, and contributors across an unprecedentedly vast spectrum of phenotypic traits. In contrast to the common approach of hand-picking variables to look for effects in a limited set of candidate measurements based on a priori research intuitions, we took a direct data driven approach capable of appreciating the full picture of the conjoint relevance of all participants and all phenotypes simultaneously. After empirically demonstrating that this deep learning approach could outperform a traditional machine learning method (i.e., PCA) in terms of its ability to faithfully capture the underlying structure of this population dataset, we used it to identify learned groupings of conjointly relevant phenotypes stratifying the family systems.

Notably, each of the four SES-centric population components uniquely married a subset of the 202 SES measures with distinct combinations of lifestyle category measures relating to cognition, mental health, environment, and demographics. In other words, different SES measures appear to be interlocked with distinct sets of phenotype categories at the population level in the ABCD Study® cohort. Hence, major forces behind population variation may be explained conjointly by an intricate web of factors with nuanced indicators of SES at the center. Component A specifically flagged SES indicators of material poverty (i.e., houses with incomplete plumbing, no telephone, and no vehicle). In contrast, unique SES measures in component D highlighted links between experiences of racial and ethnic minoritized groups, and neighborhood conditions (i.e., densely populated living). In component B, a unique set of SES indicators tracked education level (i.e., high college enrollment) and temperate climate, while E uniquely indexed families being of European ancestry as well as living in desirable and healthy environments (i.e., low commute duration, lack of pollutants in air, water, and soil). As a core conclusion from our work, among the totality of 202 SES variables, no single measure was shared amongst all 4 leading population components. This observation suggests that SES is a complex multidimensional construct that cannot be captured using only one or two of the commonly used measures of education, occupation, and yearly income.

Our findings are consistent with previous studies providing early cues that traditional monolithic indicators of SES may be insufficient when applied to more generously sampled cohorts that are closer to the heterogeneity of the population [1,42]. One such study points out that traditional socioeconomic indicators may not have the same meaning for immigrant families as for US-born families [43]. Race and ethnicity have been observed to have a complex relationship with SES, with a substantial wealth gap between races in the US at similar levels of education and occupation [44]. Additional research indicates that traditional measures of SES may overlook population-level patterns originating from distinct indicators [22,45], emphasizing the shortcomings of existing models focusing solely on these traditional measures. In line with the sociological framework of 'Intersectionality' [46], our findings indicate that markers of identity may not function independently but rather collectively influence an individual's experience of privilege or discrimination. Utilizing intersectional methodologies can reveal insights into social dynamics overlooked when investigators focus on singular identity aspects [47]. This underscores the value of our disentanglement approach, which identified distinct coherent domains among a comprehensive set of detailed SES measures, with their relation to the rest of the phenome. Reinforcing this notion, our evidence indicates that relying on overly narrow measures of SES in research could be detrimental, as this widespread current practice may obfuscate the actual contributions of wider, multifaceted SES population components [3,48].

## Executive function across SES dimensions

Within each of the 4 here discussed SES-centric components, the Neuropsychological Tests category, measuring aspects of executive function, especially working memory and attention capacity, consistently emerged as a high effect domain. Specifically, our components A, B, D, and E were all sensitive to inter-child differences in executive function through working memory task measures. Along with working memory, component E also tracked visuospatial processing flexibility and attention performance. Considering the profile of distinct SES indicators in component E, this emphasizes that the most relevant measures of executive function can differ across SES dimensions. Literature has extensively documented associations between SES and working memory performance in young adulthood [49–54]. This relationship is often identified through SES indicators of low family income or years of childhood living in poverty leading to elevated psychological stress and working memory deficits in young adulthood. In our study, census area poverty is just one of many dimensions of SES that we were able to link to executive function via our 4 SES-centric components driving population variation. Furthermore, depending on the specific SES indicators linked with executive function in each component, we identified distinct sets of co-occurring phenotype categories associated with executive function, an advantage of our multivariate approach.

## Multifaceted dimensions of deprivation: Exploring SES and beyond

As an overarching trend in our findings, our components A and D were both driven by deprivation-related SES measures. Deprivation refers to a lack of something important or something needed for good quality of life. Yet, the exact measures at play were non-overlapping as they co-occurred with different non-SES correlates across the phenome. The theme of component A, our single most explanatory component, can be summarized as relating to material poverty and its health and mental well-being correlates. Instead, our component D indexed deprivation as tracked by densely populated living and its disproportionate effects on immigrants and racial and ethnic minoritized groups.

In component A, our SES drivers related to high poverty rate and public assistance rate in census area, as well as areas scoring low on child opportunity and economic resource indices. Distinct measures of material poverty in this source of population variation included housing with incomplete plumbing, no telephone, and no vehicle, low income and low rent in area, and lack of health insurance coverage in area. In addition to SES and Neuropsychological Tests, top categories in component A were Mental Health Summary, Social Responsiveness, and Demographics. Top phenotypes, outside of SES and Neuropsychological Tests, related to mood swings, indicators of mania and depression, impulsivity, restlessness, social problems, rule breaking behavior, neighborhood risk, high screen time, and poor reasoning skills. Additionally, phenotype hits from the Mania category emerged with higher coverage (in higher proportion) and larger magnitude (effect size) in component A compared to any of our other components. These include experiencing periods of extreme happiness, intense energy, rapid thoughts, and feeling that you were a very important person.

It is generally accepted that mental health problems are distributed unevenly in our societies as a function of SES; although previous studies report mixed evidence of associations between affective disorders and SES, these problems disproportionately affect those of lower SES [55]. The behavioral and mental health indicators that emerged in our findings in association with material poverty directly reflect this established imbalance. Although using deep learning-enabled discovery in a more expansive panel of phenotypes, several of our findings are in line with earlier studies that find mental wellbeing is positively linked with SES [56,57], with less depression, anxiety, and psychosis at higher levels of SES. Arriving at the same result using different data and analytical approach speaks to the robustness of our measures on explaining population variation. Given established evidence, prolonged exposure to chronic psychosocial stressors can escalate the risk for many diseases or worsen pre-existing medical conditions such as affective disorders, cardiovascular disease, diabetes, and autoimmune disease, and reproductive inhibition [58–61]. Consequently, the appearance of these pointers to affective disorder in component A could be partly explained by the increased psychological stress associated with material poverty.

In our component A, the social behavior problems flagged are typical indicators of autism such as avoiding eye contact, rigid routines, and difficulty understanding people's tone of voice and facial expressions. The observed specific indicators of restlessness, in turn, are often associated with ADHD, including difficulty waiting turn, acts like driven by a motor, fidgets, and difficulty remaining seated. There exist several studies linking SES with ADHD [62–64] as well as autism [65,66]. Extending upon previously observed associations in alignment with our findings, impulsivity has also been noted as a feature of autism [67]. Impulsivity, along with other measures related to drivers in this component, neighborhood risk and single parenthood, have been found to relate to antisocial behavior [68]. There is similarly strong evidence in separate studies for links between other key covarying themes identified in this present component: SES and impulsivity [69–71], affective disorders and impulsivity [72–74], and SES with social abilities [75–78].

These examples demonstrate how previous studies have recognized some of the relationships identified also by our present approach. However, earlier studies fall short of providing a comprehensive understanding, as previously attempted approaches have systematically failed to reveal the intertwined nature of these association sets across disparate phenotype categories. Thanks to the multivariate nature of our analytical approach combined with the uniquely rich phenotyping in the ABCD Study®, we were able to show that many separately reported associations are in fact closely related. These associations systematically co-occur across thousands of families, thus constituting essential links of the same complex web that contributes to explaining population variation. Hence, our multi-variable approach put us in a position to identify broader dependencies between constructs belonging to phenotypic categories, typically studied in isolation, and advocates for the importance of considering each of these constructs in a study investigating any one of them.

Importantly, component A suggests that individual differences in socioeconomic measures of material poverty coincide with non-SES measures touching on executive function and reasoning ability, mental health metrics of affective disorders and impulsivity, social interaction difficulties, restlessness, rule breaking behavior, as well as demographics measures such as ethnicity and neighborhood risk. Being our component with the strongest explanatory power, the findings of component A imply that the largest portion of population variation in the USA can be explained by tight interdependencies between these sets of measures. This emphasizes the profound effect that the imbalance of economic resources probably has on health in US societies.

Also related to deprivation but from a different angle, the SES-centric component D related to low homeownership, areas with high percentile racial and ethnic minoritized groups, high population density, dense housing, and low high school graduation rates. Aside from SES and Neuropsychological Tests, top categories in component D included Diagnostic (KSADS), Demographics, and Cognitive Capacity. Top effects in these domains related to the family identifying as "Other" or "American Indian", being recent immigrants, and being Spanish speaking. Parents also identified their child as having a diagnosed phobia and often feeling fearful or anxious. Taken together, these effects can be seen as reflecting the experience and neighborhood conditions of immigrant and racial and ethnic minoritized groups. Occupying a stressful position in the social hierarchy has previously been associated with increased anxiety [33,79]. Consequently, the heightened stress stemming from the social dynamics of the immigrant experience may explain why component D exhibits drivers such as phobia, fear, and anxiety.

In agreement with our findings, previous research delivered evidence of densely populated living conditions often coinciding with immigrant status. For example, one study found that immigrants are more likely than native-born workers to work in low-wage jobs, reside in urban areas, and live in larger and more crowded households [80]. Furthermore, there is a known wealth inequity and gap in poverty rates between racial and ethnic groups in the US [81,82]. Specifically, in line with the connections identified by this component linking low rates of high school graduation, low family income, and American Indian ethnicity, it is known that American Indian and Alaska Native populations face elevated poverty rates [83] and have historically achieved lower educational outcomes [84,85] compared to the broader population. It has similarly been observed that family poverty is linked to high rates of high school non-completion for Latinos [86]. Furthermore, variations in kindergarten assessments of math and reading performance among Mexican immigrant children have been

explained by family socioeconomic circumstances, specifically relating to access to educationally enriching home experiences [87]. These examples highlight links to prior knowledge of only a fraction of the numerous relationships unveiled through our multivariate approach within this component. The top drivers identified in component D corroborate these previous findings, and expand upon them by also linking executive function, cognitive capacity, fear, anxiety, and phobia to the unique experiences of individuals as immigrants or members of racial and ethnic minoritized groups. Studies have individually associated heightened levels of fear and phobia with lower SES groups [88] and revealed imbalances across different ethnicities [89,90]. However, our present work goes beyond that previous work in scope and depth.

In summary, component D indexed deprivation from a different perspective compared to component A. That is, another dominant axis of population variation was intimately linked with densely populated living conditions that coincided in our USA-based cohort with demographic related factors such as recent immigration or belonging to a racial and ethnic minoritized group, as well as living with phobia, fear, or anxiety, and executive function and cognitive capacity. It is important to highlight that certain cognitive assessments utilized have previously exhibited cultural biases in scores, stemming from the Eurocentric nature of metrics [91,92], which could have impacted the observed race and ethnicity related associations. The interlocking relationships from this highly explanatory component D highlighted that significant differences in quality of life in the USA may be related to the language one speaks, their ethnicity, or immigrant status.

Further, components A and D both identified education as an explanatory facet of SES, but did so via unique metrics. In component A, low early childhood education enrollment emerged as a top phenotype hit, while component D tracked low high school graduation rate. Component B, another key SES-centric component, tracked a third measure of education, high college enrollment and presented yet another unique set of covarying lifestyle phenotypes. Studies do not typically compare these measures, often using just one metric of education. However, considering an array of complementary metrics capturing education enabled us to identify what previous studies were blind to: that these three distinct indicators are separate reasons behind what drives interindividual differences, each associated with distinct set of covariates. This insight tells us there is potentially more to the story than just "education" as a monolithic measure of SES, and that substituting one indicator for another or using an umbrella metric may not provide a sufficient level of granularity to draw accurate conclusions about somebody's sociodemographic profile.

### Unpacking privilege: SES drivers and distinct correlates

In contrast with SES-centric components A and D, components B and E both captured privilege-related SES measures but through different angles and with disparate sets of phenotype indicators. The theme of component B can be summarized as preferentially capturing the educational and behavioral outcomes shaped by one's upbringing. Instead, component E preferentially captured values and opportunities of those leading privileged lifestyles.

Component B was characterized by SES measures such as being in the 1st percentile of the national income, as well as child reading proficiency, high child educational opportunity via high advanced placement course enrollment and high college enrollment in census area, low social disadvantage in census area, and low rates of disability status in census area. Top categories aside from SES and Neuropsychological Tests in component B were Parent Characteristics, Social Adjustment, Diagnostic (KSADS), and Behavior. Within these covarying categories of component B were measures that relate to being non-religious, having good concentration skills, and being well behaved; as measured by the child's ability to play quietly as well as lack of lying, conduct disorder, vandalism, or bullying. Along with having good behavior, component B represented children who come from healthy home environments that promote non-violence; indicated by measures such as family members do not hit each other and do not get so angry they shout at each other.

Several studies have shown family income to be a key predictor of neglect or physical abuse [93–95]. This earlier evidence is in line with our discovered association between family SES, as characterized by being 1st percentile in national income, and living in non-violent home environments. Furthermore, evidence from studies reinforces the notion of a connection between family SES and behavioral problems in children as displayed in this component. For example, children

from low-income families are often assessed by parents and teachers as exhibiting more behavioral issues compared to their wealthier counterparts [96]. Even more specifically, studies have found that family SES is often more strongly associated with externalizing than internalizing child psychopathology and that the magnitude of this relation is likely to vary by study population and community [97].

Our results tell a slightly different story, while component B primarily flagged indicators of externalizing behavior, component A displayed indicators of both internalizing (i.e., depression, mania) and externalizing (i.e., ADHD, rule breaking) behaviors. These indicators were linked with different constellations of covarying measures, suggesting that whether family SES is more strongly associated with internalizing or externalizing behavior of the children may be complicated by a multitude of external factors. Certain studies have suggested that parental monitoring exerts a greater impact on externalizing behavior than internalizing behavior [98–100]. Consequently, it is plausible that the ties with externalizing behavior are stronger in component B because one primary driver in this component is healthy home environments, where monitoring may be more consistent. Conversely, component A may capture an influence of family SES on child behavior independent of parental monitoring levels. Hence, the outcomes derived from our multivariate approach offers important nuances to relationships identified in previous studies, underscoring the value of capturing comprehensive phenome profiles.

Prior studies show a relationship between religiosity and educational attainment [101–103]. This relationship has been found to be subtle, with evidence showing that association strength between these two constructs varies according to religion [104]. Other work has found that rather than level of religiosity, its alignment in parent and child religiosity level that predicts higher educational achievement [105]. We indeed observed high alignment between parent and child answers to questionnaire items such as the importance of following the word of god and the degree to which god is first, family second in component B. Thus, the multivariate nature of our approach appears to facilitate capturing this nuanced relationship with high-resolution.

The disability status indicator flagged in component B refers to difficulties with one or more of hearing, vision, cognition, ambulation, or self-care [106]. The association between disability status and poverty is yet another relationship that has previously been noted to be highly nuanced; the by-product of factors such as social marginalization, lack of access to education, employment, health care, and social support systems [107]. These factors are related to top drivers we uncovered in component B such as high child educational opportunity, high earning households, and households in low socially disadvantaged areas. A growing demand exists for research that delves more deeply into the intricate relationship between poverty and disability, incorporating more nuanced variables. Our study can be viewed as an initial step in addressing this call for a deeper understanding, as our approach has expanded upon the group of variables thought to be conjointly relevant in this association by bringing additional factors such as religion, home environment health, and behavioral indicators into the fold.

Summing up component B, this source of population variation suggests that SES measures of high child educational opportunity and performance, high family income, together with executive function, healthy home environments, religiosity, disability status rates, and externalizing behavioral metrics, constitute a third conjointly relevant group of variables that explain another substantial portion of population variation between families in the USA. A crucial insight gleaned from this component is that an individual's educational attainment and behavioral outcomes appear to be shaped by a spectrum of social and environmental factors more extensive than previously understood.

In component E, driving SES-related measures included proportion of European ancestry, high overall child opportunity levels, school wealth, and neighborhood wealth characterized by a low percentage of students eligible for free or reduced-price lunches. Of note, component E included distinct SES-related indicators not found in any of the other 3 components: low commute duration, low levels of pollutants, and low levels of state racism. High impact categories aside from SES and Neuropsychological Tests in component E were Demographics, Social Adjustment, Extracurricular, Deep Phenotyping Assessments, and Diagnostic (KSADS). Although component E had the most significant overlap of SES measures with component A at 25%, it still presented distinct drivers within these interlocking domains, namely, extracurricular

involvement, student gets A's in school, parent financial responsibility, having married parents, a lack of Mexican American cultural values, and low screen time. This result adds another brick to the notion of SES as a multidimensional construct to identify relevant covarying factors of interest.

The Extracurricular category is a unique driving factor distinguishing component E from our other modes of population variation, with measures being represented in higher proportion and with larger effect size in component E compared to any other component. This category captured child participation in activities like reading, sports, and playing musical instruments. Our findings were mostly confirmatory of existing literature, which has established a relationship between SES and extracurricular activity participation. Specifically, a previous study identified family affluence as a predictor of organized sport participation [108]. The family affluence scale in said study is based on material predictors, more like those of component A, such as how many computers a family owns and owning a vehicle. Conversely, in component E, we identified SES indicators related to a child's environment, such as school and neighborhood wealth, as key to profiling the relationship between extracurriculars and SES. These findings imply that incorporating additional SES measures pertaining to a child's environment could prove advantageous for future studies focused on implications of extracurricular participation. The relationship between organized activity participation and emotional wellbeing is one that has been observed to depend on the ethnicity and socioeconomic status of adolescents [109], and could thus benefit from increased socio-environmental context. There is a wealth of studies focused on the implications of extracurricular participation in other areas such as improving psychological wellbeing, reducing risk behaviors, improving academic outcomes, and increasing social capital [110–114] where our findings could prove similarly beneficial.

The emergence of Mexican American cultural values in component E adds more color to the already vivid depiction of disadvantage that racial and ethnic minoritized groups experience in the US. Rather than being tracked by indicators of native language spoken or recent immigration status like component D, this component presented low scores on parent questionnaire items relating to cultural values of ethnic minoritized groups such as religiosity, family obligation, being self-reliant, caring for older relatives, and parent sacrifice to ensure a better life for one's child. These findings corroborate literature that highlights a link between youth of Mexican origin and socioeconomic status via family and neighborhood poverty [115,116]. Crucially, we observed a distinct array of interrelated themes when capturing the racial and ethnic minoritized experience through cultural values in component E. In component E, extracurricular participation, school, and neighborhood wealth emerged as pivotal co-occurring themes, while in component D, covarying measures included cognitive capacity, phobia, fear, and anxiety. Furthermore, these two components delineate the experiences of different racial and ethnic groups, with component E specifically identifying individuals of European descent. Hence, our findings indicate that distinct background variables may be more important than others depending on the specific racial and ethnic group under study and the measures employed to characterize that group.

The married parents' indicator in component E stands in contrast to component A being driven by single parent households, and component D being associated with parent living with partner. Literature has linked parental relationship status to a host of child outcomes similar to those identified in our components such as executive function, mental health, poverty, higher delinquency, and worse educational achievement. One study found that children from low SES families living with one parent demonstrated lower performance on executive function tests when compared to their counterparts from similarly low SES backgrounds who lived with both parents [117]. Another observed that adolescents in married, two-biological-parent families generally fare better than children in any of single-mother, cohabiting stepfather, and married stepfather families in terms of academic and behavioral indicators [118]. The findings across components A, D, and E confirm and expand upon the array of life outcomes linked to parental relationship status. Notably, our framework captured the nuance of this relationship by attributing different covarying factors of interest depending on the status of a child's parental relationship.

Another finding of interest is the emergence of low screen time in component E contrasting with high screen time in component A. Higher screen time has been shown to be associated with lower socioeconomic status and belonging to

a racial and ethnic minoritized group [119–122]. Component E indexed European ancestry while A captured material poverty, so the respective associations with screen time are in alignment with literature findings. Our analytical approach adds value here by identifying different covarying factors of interest depending on amount of child screen time. This may add additional color to the implications of screen time. For example, component A could imply an association between high screen time and affective disorders such as bipolar disorder, mania, and depression, as well as ADHD-typical and autism-typical behavioral traits.

To summarize, component E offers a complementary perspective on privilege in contrast to component B, suggesting that measures of ancestry, executive function, family values, neighborhood desirability, and lifestyle are together relevant in explaining another considerable portion of population variation in families in the USA. In particular, component E spotlights the noteworthy disparities in lifestyle and opportunities influenced by sociodemographic group and environmental factors.

## Rethinking population stratification: Current knowledge and future implications

As illustrated, literature supports many associations uncovered via our deep learning approach, but in most cases the previously reported relationships have been established on a pairwise or variable-variable basis. Identifying that each of these previously identified pairwise relations in fact belong to distinct covarying groupings that underpin major sources of population stratification represents a novel finding uniquely enabled by the multivariate approach and population scale data used in this study. It is important to note that our multivariate approach offers, probably more realistic, interpretation at the level of groups of variables identified, without presupposing that specific pairs of single individual variables tell the whole story [123,124].

Identification of the distinct covarying phenotype groupings of this investigation was made possible by the granular measures of SES available in the ABCD Study® beyond the typical education, income, and occupation. SES measure definitions utilized in prior studies are seldom as granular as the phenotype hits identified via our present framework. As our investigation has illustrated, even slight modifications to the SES construct can yield a different spectrum of pertinent covarying sources of interindividual differences. Consequently, our findings underscore the importance of conceptualizing SES as a multidimensional construct, more so than it has been in the past. This bears particular importance when selecting sources of background variation to take into account when studying a phenotype of interest to avoid spurious associations.

Our findings also suggest that wellbeing later in life may be strongly influenced by factors that we can reliably measure in a large cohort at adolescence. For example, screen time, rarely available in previously existing population cohorts, which emerged as a driving covariate alongside SES in components A and E of our study, has been linked with two factors identified as relevant for wellbeing later in life; physical activity levels [119] and sleep quality [125,126]. The latter is significant for its various health benefits such as hormonal balance and cognitive function. Lack of sleep is also related to worse mental health outcomes [127–129] and higher stress levels [130–132] two additional factors crucial for longevity, or what has been termed "health span" – the period of life spent in good health, free from chronic diseases and debilitating conditions [133]. Here, we identified an intricate link between specific mental health indicators and SES measures in component A. Low SES is in turn known to be linked with chronically elevated stress levels [3,50,134]. These connections culminate in an interlocking relationship between specific measures of SES in our study and four general factors of health span: stress, sleep, mental health, and physical activity. While much current research on health span primarily focuses on well-being in later stages of life, our findings imply that distinct SES dimensions may be related the health trajectory of children at an earlier stage than conventionally anticipated.

Although not directly comparable to our present study due to their considerably lower number of variables considered, scale of data resource, and embraced analytical approach, earlier multivariate association-based studies focused on population stratification have identified certain groupings of covarying phenotypes driving population variation similar to some

of ours. Measures relating to poverty, parent unemployment, education level, and cognitive ability, similar to our present hits in component A, were identified in a predominant mode of covariation by a study utilizing an earlier release of the ABCD Study® to identify multivariate brain-behavior mappings from 617 variables [135]. Additionally, a study performed exploratory factor analysis on 2010 census-based American Community Survey (ACS) using 13 variables to characterize the contribution of one's environment to explaining differences in neurocognition [54]: these authors found key factors relating to neighborhood SES, household composition, and language which are very similar to drivers identified in component D of our analysis. Illustrating the power of variables driving population variation to help uncover critical differences in brain function, participant scores on the SES factor identified by this study were subsequently demonstrated to be associated with differential activation amplitudes across the sensorimotor-axis of the brain [136].

In another study, which applied principal component analysis to 15 US census data variables, the most explanatory component was explained by factors such as impoverished living conditions, low high school graduation rate, minoritized status, and single parent households [22]. These factors reflect a combination of the driving effects from components A and D in our study. Furthermore, the most prominent profile of a study investigating multivariate relationships between 128 urban environmental metrics and 21 psychiatric symptoms in the UK Biobank [137] was also composed of an aggregation of drivers observed across components A and D in our study. Specifically, high levels of deprivation and air pollution, increased affective disorder symptoms, as well as poor, dense inner-city neighborhoods were noteworthy commonalities. In our study, we leveraged a broader range of phenotypes, a factor that likely empowered us to carefully disentangle components A and D which were conflated in this previous research.

The findings in our study also related to a study which uncovered key correlates of the Covid-19 pandemic experience [138]. Themes such as marital status, household income, immigration status, language preference, parental employment, and preference language spoken in everyday life identified as key to differentiating a family's pandemic experience were also uncovered in our investigation as important drivers of interfamily differences that explain a notable portion of population variation in the USA. This result implies that phenotypes exhibiting the greatest variations in society are also responsible for the greatest divergence of experiences during a major population-level public health event such as a pandemic. The agreement between our study and similar studies provides reassurance regarding the robustness and generalizability of our findings.

More broadly, the drivers outlined in the four distinct SES-centric components identified show a certain alignment with factors of population diversity defined by the WHO as "social determinants of health" (SDOH) [23]. As outlined in the introduction, SDOH are receiving increasing attention as potential root causes of population health inequalities [24,25]. Included under the broad umbrella of SDOH are psychosocial factors such as subjective perception of social status [37]. There exists compelling evidence indicating perceived social disadvantage as a fundamental factor contributing to the association between lower SES and unfavorable health outcomes [28,30–32,35,36]. Coupling this with the notably higher poverty rates and income inequality in the US compared to other high-income nations [24], it is reasonable to assert that the ABCD Study® cohort is primed for SDOH to play a pivotal role as primary drivers of interfamily differences.

Our collective findings can help serve as a guide for future investigations on relevant covarying factors of interindividual differences to be included when investigating a phenotype of interest. To further evaluate the robustness of drivers unveiled in our analyses, we encourage future investigations into major sources of interindividual differences leveraging population scale cohorts from different regions. This could serve to both help identify sources of population stratification pervasive across broader geographies as well as provide insight into relationships identified in our US cohort that are region specific. This contextualization would allow discerning with a broader scope the relative significance of specific sources of population stratification, enhancing the actionable insights for guiding future investigations.

Beyond this, future research could further contextualize the relationships identified in our study by integrating genetic data and comparing our findings with other intersectional frameworks. Incorporating genetic influences would help disentangle inherited and environmental contributions to interindividual variability, while comparing our results with other

intersectional models could provide deeper insights into how SES, race, and gender interact to shape phenotypic outcomes. These extensions would enrich our understanding of the complex drivers of interindividual differences highlighted in this study.

Identifying key sources of interindividual differences is just the beginning, the end goal of improving modeling performance for heterogenous population scale data will require architectures shifts to incorporate these identified strata. Early candidate approaches include Bayesian hierarchical modeling [139], multilevel regression and post-stratification [140], and propensity score matching [141]. By evaluating such methods, we can advance toward diversity-aware modeling, ensuring that findings are applicable across a wide spectrum of subgroups within the broader population.

In conclusion, our study indicates that interindividual differences stem from a complex network of Social Determinants of Health (SDOH), with specific dimensions of socioeconomic status (SES) at their core. These findings underscore the critical need to incorporate SDOH in biomedical research, and we aspire for our results to encourage the increased integration of SDOH in future studies.

## Methods

### ABCD Study® population data resource

The Adolescent Brain Cognitive Development<sup>SM</sup> Study (ABCD Study®) is a longitudinal population-based cohort – the largest biomedical child development study of its kind [20]. We obtained the entirety of the ABCD Study® Data Release 4.0 tabulated phenotypic data (including behavioral, clinical, cognitive, and sociodemographic data). This data release included data from 11,875 youth from 21 sites around the United States from ages 9–10 into young adulthood. Sample demographics include 47.8% female; 52.1% White, 15.0% Black, 20.3% Hispanic, 2.1% Asian, and 10.5% Other [142]. Other includes participants of Native Hawaiian, Pacific Islander, Alaskan Native, American Indian and multiple races [143]. From the ABCD Study® data release 4.0 we analyzed data from the baseline, screener, 6-month, and 1-year events because these included data for >95% of families present at the time of baseline assessment. All protocols for the ABCD Study® are approved by either a central or site-specific institutional review board committee [144]. Caregivers have provided written, informed consent and children provide verbal consent to all research protocols [145]. This dataset is administered by the National Institutes of Mental Health Data Archive and is freely available to all qualified researchers upon submission of an access request. All relevant instructions to obtain the data can be found online (https://nda.nih.gov/abcd/request-access).

The ABCD Study® utilized a multi-stage probability sampling strategy as an effort aimed at a cohort that would be close to representative of the sociodemographic composition of the general US population [143]. With the goal of minimizing systematic sampling biases, a stratified probability sampling process was utilized to ensure randomization and representativeness of selected schools across the US. Due to constraints on research expertise and the neuroimaging equipment requirements, the 21 collaborating sites were more likely to be located in urban areas, which resulted in a relative under-representation of rural youth. Thus, although the ABCD Study® has made efforts to approximate the diversity of the US population on sex, race/ethnicity, and socioeconomic status, the degree to which this participant sample is representative of the U.S. population may vary across outcome measure examined [146].

### Data curation protocol

Our analysis included a comprehensive array of behavioral, clinical, cognitive, and sociodemographic candidate phenotypes, obtained from the ABCD Study® 4.0 Data Release. Tabulated data including 11,875 participants with 13,638 phenotypic variables available were imported and processed using the Python data science stack.

Pruning was performed to retain only the phenotypes with data populated for at least 80% of the total participants. To deal with nearly-constant data columns, any phenotype with an occurrence of >99% for the most common compared to

the second most common value was removed to exclude columns with near-zero variance. Values indicating participant non-compliance were processed as NaN. Data processing workflows were separated for discrete and continuous variables for imputation of missing values and further curation. Discrete data columns were one-hot encoded. For example, the variable "ethnicity" with categories White, Black, Hispanic, Asian, and Other was transformed into five binary variables: "White" represented as [1, 0, 0, 0, 0], "Black" as [0, 1, 0, 0, 0], "Hispanic" as [0, 0, 1, 0, 0], "Asian" as [0, 0, 0, 1, 0], and "Other" as [0, 0, 0, 0, 1]. Continuous data columns underwent robust z-scoring similar to data curation protocols of previous research [135]. Values greater than four standard deviations away from the mean in magnitude ($z > 4$) were replaced with the largest magnitude value within four standard deviations of the mean (sometimes called winsorization). This step enabled us to avoid adverse effects on the quality of our model fits due to outliers while also retaining as many phenotype measures as possible. Finally, to ensure our analyses were conducted on an equally complete sample across data collection timepoints, only collection events which included >95% of the number of families present at the time of baseline assessment were retained.

Our resulting data considered for further analysis consisted of 11,875 participants each with 8,902 features spanning the baseline, screener, 6-month, and 1-year event data collection timepoints. The NIMH Data Archive API (https://nda.nih.gov/nda/apis) was used to retrieve ABCD Study®-linked categories for each phenotype. The complete repertoire of 8,902 curated target phenotypes spanned 23 predefined categories and provided the basis for all downstream analyses. The complete list of phenotypes with category and description is available in the code repository https://github.com/dblabs-mcgill-mila/abcd_dl_analysis.

## Design of deep learning model architecture

Our analysis utilized a recently emerged deep learning framework, the conditional variational autoencoder (CVAE) [147], which has been shown before to be effective in biomedical data [148]. We adapted the architecture utilized in the PyTorch implementation of the CVAE from previous work (https://github.com/theislab/trvaep, as utilized in Lotfollahi et al., 2020) to create a modeling framework directly tailored to the structure of our data resource. Site is a known important factor of variation in the data [1], so in this analysis we elected to use a CVAE for its ability to learn complex relationships between input variables while explicitly acknowledging site-related variation in our phenome variables during model development. Specifically, deep autoencoder neural network architectures have been shown to outperform ordinary principal component analysis (PCA) [21] on tasks such as facial and handwritten digit recognition [149,150]. In a population genetic variation study, VAEs demonstrated the ability to capture a comparable level of complex population structure using fewer latent dimensions (i.e., components) than PCA [151]. Here, as a technical preliminary step, we aimed to validate the CVAE efficacy by showing that it can outperform PCA in its representation of the ABCD Study® data.

Data were randomly split into training, validation, and test splits using StratifiedShuffleSplit from the scikit-learn package. This cross-validation scheme segments data into randomized folds while preserving the percentage of samples for each class. We used 80% of the participants as a training split (n = 9511), 10% as a validation split (n = 1176), and 10% as a test split (n = 1188). This data splitting scheme ensured there were an equal proportion of participants from each of the 21 ABCD Study® sites in each dataset split.

Using these dataset splits, we evaluated a collection of CVAE architectures. We trained the model parameters of each architecture using a variant of stochastic gradient descent solvers, namely the Adam optimizer, and with early stopping as a form of regularization [152]. Nonlinear ReLU activation functions were used in encoder and decoder hidden layers, and linear activation functions on the decoder output layer to enable both positive and negative outputs. Each architecture was trained on train split participant sample, used the validation split sample for early stopping, and was evaluated via mean squared error (MSE) reconstruction loss on the test split sample. Based on prior exploratory data analysis using PCA via the scikit-learn package with a 100-dimensional latent space (i.e., 100 components), we identified via the widespread elbow criterion that 10 components account for a high proportion of variance in the data compared to the remainder of

the top 100 components (S1 Fig in S1 File). To enable comparison between PCA and CVAE with equivalent latent space dimensionality and a balance between model simplicity and representational capacity, we opted for 100-dimensional latent space in all explored CVAE architecture variants. As a deep learning approach optimized through stochastic gradient descent solvers, the CVAE optimization process is non-convex. Initial weight values of nodes in the neural network can lead the model to converge to different local minima. Thus, to robustly characterize the performance of our model we trained 25 instances of six different CVAE architectures with different randomized model weight initializations in each instance (Table 2). This allowed us to confidently select the CVAE architecture exhibiting the lowest average MSE reconstruction error ('loss') on the test set across 25 different weight initialization random seed values.

All examined deep learning architectures utilized identical two-layer encoder and decoder networks, where we index the depth of the network as the number of layers of learnable weights [153]. The differences between the six evaluated candidate architectures were in the width of the hidden layer and the preprocessing of data prior to input to the CVAE. Architectures 1, 2, and 3 used hidden layer widths of 200, 400, and 800 neurons and ingested the data directly as output from preprocessing. In CVAE architectures 4, 5, and 6, prior to being ingested by the CVAE, each phenotypic measurement was submitted to PCA to increase the robustness of subsequent steps by avoiding potential issues with rank deficiency and overfitting to noise. Applying PCA to the input data also enabled providing the CVAE with coherently represented input dimensions, abstracting across the different variable representations (i.e., continuous and discrete) that naturally occur in population phenomics in our dataset. PCA was applied to the input data to reduce the dimensionality to 400, 200, and 150 features before feeding into CVAE models with hidden layer widths of 200, 150, and 125 respectively in architectures 4, 5, and 6.

We empirically compared the performance of these six candidate CVAE architectures by their ability to successfully reconstruct the unseen test split sample versus that of PCA as a baseline model. For PCA, because the input data were larger than 500x500 and the number of components to extract was lower than 80% of the smallest dimension of the data, the 'randomized' SVD solver method was enabled (sci-kit learn function). In so doing, a random state value must be passed to PCA for reproducibility. PCA was fit on the same training dataset as the CVAE and MSE reconstruction performance was calculated on the test split data for 25 iterations of solver random state to identify the range of possible MSE reconstruction performance values produced by PCA. Mean and standard deviation of MSE reconstruction performance were computed across iterations and compared to that of our six candidate CVAE models (S2 Fig in S1 File).

Based on this thorough examination across these choices, we identified the best performing architecture, as characterized by best average MSE reconstruction performance on the untouched test split sample, as architecture 5 which first extracts the top 200 PCA components to feed into our CVAE with 150-dimensional encoder and decoder hidden layers. The CVAE mean reconstruction performance (mean MSE = 865.9, SD = 1.09) of architecture 5, the best performing model architecture, exceeded the reconstruction performance of PCA (mean MSE = 871.9, SD = 0.20) by more than 2 standard deviations. Hence, this deep learning architecture was utilized for all subsequent analysis steps.

**Table 2. Evaluated CVAE architectures. Each architecture was trained with different randomized model weight initializations across 25 instances to compute the mean MSE reconstruction error.**

| CVAE Architecture | Input PCA dimensionality | Hidden layer dimension | Mean MSE reconstruction loss | Standard deviation |
|---|---|---|---|---|
| 1 | N/A | 200 | 966.7 | 19.60 |
| 2 | N/A | 400 | 882.3 | 2.53 |
| 3 | N/A | 800 | 902.9 | 4.58 |
| 4 | 400 | 200 | 867.4 | 1.48 |
| **5** | **200** | **150** | **865.9** | **1.09** |
| 6 | 150 | 125 | 866.1 | 0.50 |

## Design of optimization objective

The variational autoencoder (VAE) [154] is a deep conditional generative model which aims to maximize the following likelihood function:

$$p_\theta (X|C) = \int p_\theta (X|Z, C) \, p_\theta (Z|C) \, dZ$$

Here $X$ denotes a high dimensional random variable. We are aiming to maximize the probability of each observation $x_i$ from $X$ under the assumed data-generative process. In our case, $X$ is the training split sample. C is a random variable representing conditions. The case in which $C$ is not null is known as the conditional variational autoencoder (CVAE) [147] and is an extension of the original VAE framework. For the purpose of the present study, we conditioned on the participant's ABCD Study® data collection site to inform the deep neural network of this known important factor of variation, as it derived low-rank factors of variation that best explain variation across the totality of thousands of input phenotype variables (cf. above). $p_\theta$ denotes a probability distribution parameterized by $\theta$, which corresponds to the decoder parameterized by a deep neural network itself. $Z$ is a latent variable assuming a multivariate Gaussian distribution as a prior.

The key idea of the VAE is to draw samples $z_i$ from $Z$ that are likely to produce the observed values $x_i$ from $X$. To do so, the intractable posterior distribution $p_\theta (Z|X, C)$ is approximated using the variational distribution $q_\phi (Z|X, C)$. $q_\phi$ denotes a probability distribution parameterized by $\phi$, which corresponds to the encoder neural network. As is standard in VAEs, $q_\phi$ is parameterized as a Gaussian distribution, that is, $q_\phi (Z|X, C) = \mathcal{N} (Z|\mu(X, C), \sigma(X, C))$, where the location parameter $\mu$ and the scale parameter $\sigma$ are implemented via the encoder neural network. The encoder distribution $q_\phi (Z|X, C)$ is used to assign probability mass to values of $Z$ that are likely to produce actually observed values of $X$. The decoder distribution $p_\theta (X|Z, C)$ is the output distribution that is produced by sampling from $Z$ to reconstruct the original ambient space $X$. In other words, $q_\phi$ is encoding $X$ into $Z$ and $p_\theta$ is decoding $Z$ to reconstruct $X$. The objective function used to optimize the CVAE in practice, $L_{CVAE}(\theta, \phi)$, can be obtained from the distance between true posterior $p_\theta (Z|X, C)$ and the encoder distribution $q_\phi (Z|X, C)$, this is known as Kullback–Leibler (KL) divergence:

$$KL \left[ q_\phi (Z|X, C) \, \| \, p_\theta (Z|X, C) \right] = E_{q_\phi(Z|X,C)} \left[ q_\phi (Z|X, C) - p_\theta (Z|X, C) \right]$$

By reordering certain terms and leveraging the definition of KL divergence, the encoder distribution $q_\phi$ can be linked to the decoder distribution $p_\theta$ as follows:

$$log p_\theta (X|C) - KL \left[ q_\phi (Z|X, C) \, \| \, p_\theta (Z|X, C) \right] = E_{q_\phi(Z|X,C)} \left[ log p_\theta (X|Z, C) \right] - KL \left[ q_\phi (Z|X, C) \, \| \, p_\theta (Z|C) \right]$$

The left side describes the log likelihood of the data denoted by $\log p_\theta (X|C)$ (the quantity that we want to maximize) and an error term that depends on the capacity of the model. This error term ensures $q_\phi$ is as complex as $p_\theta$ and will approach zero with a high-capacity model [155]. Therefore, we directly optimized $\log p_\theta (X|C)$, which is known as the variational lower bound (often referred to as the evidence lower bound, abbreviated ELBO). Optimizing the ELBO facilitates the concurrent fitting of both the encoder parameters $\phi$ and the decoder parameters $\theta$. This provides us with the following objective function:

$$L_{CVAE}(\theta, \phi) = E_{q_\phi(Z|X,C)} \left[ \log p_\theta (X|Z, C) \right] - KL \left[ q_\phi (Z|X, C) \, \| \, p_\theta (Z|C) \right]$$

The first term in the objective function, $L_{CVAE}(\theta, \phi)$, is a reconstruction error term which encourages the decoder to learn to reconstruct the data when using samples from the latent distribution. The second term is the KL divergence between the encoder distribution $q_\phi$ and the multivariate Gaussian prior distribution for $Z$. This term encourages the encoder distribution to be normally distributed, regularizing the latent space.

### Quantifying the role of candidate phenotypes in each derived CVAE component

To estimate the contribution of each individual phenotype to the extracted latent variables learned by the CVAE (referred to as components), we computed phenotype loadings. These were obtained for each component by computing the Pearson's correlation coefficients between respective CVAE component scores and the original phenotype measurement across participants in the test split data. Thus, loadings quantify the strength and direction of the relationship between the original phenotype and the identified CVAE components across cohort participants. Stronger loading values represent greater importance of contribution to the structure of the learned components, offering insights into which phenotypes drive the identified patterns of population variation. To determine the most important phenotypes driving coherent population variation as captured by each component, we used loading magnitude as a metric which we refer to as 'weight strength'.

### Identifying most explanatory components

Unlike ordinary PCA, the CVAE does not naturally rank components in order of explained variance. For this reason, we have determined this post-hoc after fitting the CVAE parameters. While there are several methods that could be used to quantify the variance explained by each component, we decided to use an approach of iteratively zeroing out participant-wise expressions (scores) in all but one of the 100 CVAE components of the latent variable $Z$ and using this alone to reconstruct the test split data. We selected this ("ablation") method because it directly measures the impact of each component on MSE reconstruction performance in isolation (within the context of a multivariate latent factor model). In other words, it measures the importance of the underlying patterns captured by each component in explaining the overarching structure and variation of our population data. Using this approach, the highest-ranking components in terms of variance explained were those which had the least degradation in MSE reconstruction performance compared to using all 100 components. That is, the latent space components were ranked in terms of best individual MSE reconstruction performance on the test split data. By charting the ranked contribution of each component, we identified a visual 'elbow' indicating that the top 10 ranked components explained a high proportion of variance in the data compared to the subsequent 90 components (S3 Fig in S1 File). This metric allowed us to focus on these 10 most explanatory components for the sake of interpretation of key drivers of population variation. Pearson's correlation was applied to understand the degree of difference in learned structure between top 10 ranked CVAE and PCA components (S4 Fig in S1 File).

### Isolating top component phenotypes based on percentile thresholding using weight strength

Our goal was to identify key drivers among the phenotypes flagged by the discovered components of major population stratification that track interfamily differences. With 8,902 total phenotypes considered, we needed to identify which phenotypes were most important or most characteristic for a component at hand, relative to the other components derived by the CVAE. For this purpose, we applied thresholding to retain phenotypes only in the 5 components among all 100 components (95th percentile) where they exhibited the highest weight strength. That is, for each phenotype we ranked all 100 components by weight strength and retained each phenotype only in its top 5 ranked components. Applying this metric enabled us to reduce the number of variables to only those most important for characterizing each component. Calculation of phenotype weight strength is outlined above.

### Characterizing components by driving predefined categories

All phenotypes included in our study belonged to one of 23 predefined categories: these are linked at the level of ABCD Study® data collection instrument (i.e., study, assessment) as part of the NIMH Data Archive (NDA) data structure. We used the NIMH Data Archive API (https://nda.nih.gov/nda/apis) to retrieve ABCD Study®-linked categories for each phenotype, which led to our variables being naturally grouped into 23 predefined categories. Some categories were renamed to

describe the encompassed variables more appropriately (see here for original and updated category names: S3 Table in S1 File). We used these categories, covering health and wellness, cognition, behavioral and psychosocial factors, as well as personal and lifestyle profiles as a way of characterizing driving themes for each component at a high level for each of our top 10 ranked CVAE components. After thresholding phenotypes by weight strength across components (cf. above), phenotypes were grouped into these 23 predefined categories for each of the top 10 components (called A-J for convenience). Driving categories per component were obtained by calculating sum of weight strengths for all retained phenotypes in a category by the count of retained phenotypes in that category, producing a mean weight strength per category in each component. Instead of only studying and interpreting single phenotype hits, this metric enabled us to consider the top driving ABCD Study® phenome categories that characterize each component as a complementary view of our results.

### US state out-of-sample prediction workflow

To test whether specific US states from the ABCD Study® cohort exhibit specific SES patterns, we carried out analyses predicting participants' US state residency utilizing participant SES phenotype scores from our four identified SES-centric components (A, B, D, and E). We computed participant scores in each of the four SES-centric components based solely on their measurements in Socioeconomic category phenotypes. These scores were computed as the dot product between a participant's original SES phenotype measurements and SES phenotype loadings in each component (cf. above for explanation of loadings computation). We used these SES phenotype-only component scores as input features to train a logistic regression classifier using each participant's ABCD Study® data collection site-linked state as the labels.

We used logistic regression (as implemented in scikit-learn) trained with 10-fold cross validation using all 11,875 ABCD Study® participants. Using one-versus-rest logistic regression, a separate binary logistic regression classifier was trained for each of the 17 state labels represented in the ABCD Study® data. The goal was to distinguish between participants who reside in a particular state and those who reside in any other state. Data were partitioned into 10 equally sized subsets and the model was evaluated 10 times, each time using a different fold as the test split sample and the remaining 9 folds as the training split data. This approach allowed us to obtain a more reliable estimate of the model's performance. The 10-fold cross validation process outputs a state prediction for each of the 11,875 participants. From the participant state predictions, we generated a confusion matrix (17 by 17) reporting the percentage of times state label was correctly predicted across participants (diagonal) versus percentage of incorrect classifications potentially systematically skewed towards other states (off-diagonal) (S5 Fig in S1 File).

The cross-validation cycle produced 17 binary classifiers in each fold, each associated with one of the states and interpretable in terms of its component-specific predictive coefficients. For each of these state classifiers, we take the mean of the coefficient weights across folds to obtain the average predictive strength of each feature (SES phenotype component score) (S6 Fig in S1 File). This tells us which component is most influential in predicting a participant's residency in that particular state. Along with identifying the most predictive component per state, we also computed kernel density estimate (KDE) plots for each state to visualize the distribution of participant scores in each of the 4 components for each state. These plots, analogous to histograms but using a continuous probability density curve, enabled us to identify which states have the most distinct SES signatures across our 4 SES-centric components.

### Code availability

The processing scripts and custom analysis software used in this work are available in a publicly accessible GitHub repository, along with examples of key visualizations in the paper: https://github.com/dblabs-mcgill-mila/abcd_dl_analysis.

### Materials & Correspondence

Correspondence and material requests should be addressed to Danilo Bzdok.

## Supporting information

**S1 File. S1 Fig. The first 10 PCA components capture a high proportion of variance in the ABCD Study®.** a) Individual component variance explained. b) Cumulative variance explained across components. **S2 Fig. Best CVAE architecture outperforms PCA on reconstruction loss.** We compared the mean squared error (MSE) reconstruction loss on the same test split data across 25 different random seed values for PCA solver and CVAE model weight initializations for 6 different CVAE architectures with different hidden layer sizes, all with a 100 dimensional latent space. CVAE architectures 1, 2, and 3 used hidden layer sizes of 200, 400, and 800 respectively. CVAE architectures 4, 5, and 6 first utilized PCA to reduce the data to dimensionality 400, 200, and 150 respectively and had respective hidden layer sizes of 200, 150, and 125. **S3 Fig. The top 10 ranked CVAE components account for a high proportion of explained variance.** Plotting the CVAE components in order of variance explained lead to the observation that ten components account for a high proportion of variance in the data, indicating that key modes of population stratification exist. **S4 Fig. Structure captured by CVAE components predominantly distinct from PCA components.** Pearson's correlation was applied to the top 10 components for both CVAE and PCA ranked by explained variance. The highest correlation is observed between the top PCA component and a range of CVAE components. **S5 Fig. Identified SES-centric components predict participant state residency.** Confusion matrix reporting the percentage of times state label was correctly predicted across participants (diagonal) versus percentage of incorrect classifications to other states (off-diagonal). **S6 Fig. Different SES components are more influential in predicting residency in different US states.** Mean logistic regression coefficient values across 10-fold cross validation for each state one-versus-rest classifier.
(ZIP)

## Author contributions

**Conceptualization:** Justin Marotta, Danilo Bzdok.

**Data curation:** Justin Marotta.

**Formal analysis:** Justin Marotta.

**Investigation:** Justin Marotta.

**Methodology:** Justin Marotta.

**Software:** Justin Marotta.

**Supervision:** Danilo Bzdok.

**Validation:** Justin Marotta, Danilo Bzdok.

**Visualization:** Justin Marotta.

**Writing – original draft:** Justin Marotta.

**Writing – review & editing:** Shambhavi Aggarwal, Nicole Osayande, Karin Saltoun, Jakub Kopal, Avram J Holmes, Sarah W. Yip, Danilo Bzdok.

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
