## [Decision Letter · Decision Letter 0]

Dear Dr. Marotta,

Thank you for submitting your manuscript to PLOS ONE. After careful consideration, we feel that it has merit but does not fully meet PLOS ONE’s publication criteria as it currently stands. Therefore, we invite you to submit a revised version of the manuscript that addresses the points raised during the review process.

We look forward to receiving your revised manuscript.

Kind regards,

Arnold Adimabua Ojugo, PhD

Academic Editor

PLOS ONE

Journal Requirements:

“DB is a shareholder and advisory board member at MindState Design Labs, USA.”

4. We note that Figure 8 in your submission contain map/satellite images which may be copyrighted. All PLOS content is published under the Creative Commons Attribution License (CC BY 4.0), which means that the manuscript, images, and Supporting Information files will be freely available online, and any third party is permitted to access, download, copy, distribute, and use these materials in any way, even commercially, with proper attribution. For these reasons, we cannot publish previously copyrighted maps or satellite images created using proprietary data, such as Google software (Google Maps, Street View, and Earth). For more information, see our copyright guidelines: http://journals.plos.org/plosone/s/licenses-and-copyright.

 a. You may seek permission from the original copyright holder of Figure 8 to publish the content specifically under the CC BY 4.0 license. 

Reviewers' comments:

Reviewer's Responses to Questions

**Comments to the Author**

1. Is the manuscript technically sound, and do the data support the conclusions?

Reviewer #1: Yes

Reviewer #2: Yes

2. Has the statistical analysis been performed appropriately and rigorously?

Reviewer #1: Yes

Reviewer #2: Yes

3. Have the authors made all data underlying the findings in their manuscript fully available?

Reviewer #1: Yes

Reviewer #2: Yes

4. Is the manuscript presented in an intelligible fashion and written in standard English?

Reviewer #1: Yes

Reviewer #2: Yes

Reviewer #1: The paper emphasizes that social stratifications, particularly those linked to SES, should be understood as multidimensional and interconnected with broader social determinants of health. The study encourages considering the full spectrum of SES when studying population health and behavior. All data were made available and well defined in the body of the paper.

Reviewer #2: 1) the manuscript is technically sound and the data support the conclusion because the study identified the key drivers among the phenotypes flagged by the discovered components of major population stratification which tracks the interfamily differences.

2) the authors made all data and data source underlying the findings in their manuscript fully available.

3) the statistical analysis has been performed appropriately and rigorously and comparing ordinary PCA with CVAE component.

4) the manuscript was presented in an intelligible fashion and written in standard English, however unnecessary repetitions should be avoided.

The figures and tables are placed faraway from the discussions. Reduction in the paper size would make room for the avoidance of repetitions seen across the study discussions. Thereby enhancing the quality and understanding of the study.

**Do you want your identity to be public for this peer review?** For information about this choice, including consent withdrawal, please see our Privacy Policy

Reviewer #1: No

Reviewer #2: No

---

## [Author Response · Author response to Decision Letter 1]

27 Mar 2025

Reviewer comments:

Although the study placed a lot of emphasis on SES, it might not have taken into consideration other important variables, like genetic impacts or the intricacy of intersectionality (the ways that race, gender, and SES intersect in particular ways).

We appreciate the reviewer’s insightful comment. We would like to reiterate that our study did not a priori focus on SES but rather conducted a data-driven phenome-wide analysis across more than 10,000 variables in the ABCD Study®. In a bottom up fashion, SES emerged as a key factor through this approach, rather than being selected in advance. To our knowledge, this is the largest study of its kind to comprehensively examine the totality of behavioral information in this manner.

Additionally, our use of a variational autoencoder (VAE) as a deep learning approach inherently captures non-linear interactions between all variables in the phenome, as captured by the ABCD cohort. The different components of the VAE encode meaningful latent factors that reflect population-level variation, some of which naturally align with intersectional dimensions such as the combined effects of SES, race, and gender. In this way, our approach does capture the complexity of intersectionality in an implicit and data-driven manner.

While genetic factors and intersectionality are undoubtedly important, our approach was designed to identify the strongest associations present in the data without imposing specific hypotheses. That said, we acknowledge that further work integrating genetic data and intersectional frameworks could build on our findings, and we now briefly discuss this in the revised manuscript (see page 40):

"Beyond this, future research could further contextualize the relationships identified in our study by integrating genetic data and comparing our findings with other intersectional frameworks. Incorporating genetic influences would help disentangle inherited and environmental contributions to interindividual variability, while comparing our results with other intersectional models could provide deeper insights into how SES, race, and gender interact to shape phenotypic outcomes. These extensions would enrich our understanding of the complex drivers of interindividual differences highlighted in this study."

We appreciate the reviewer’s suggestion, as it helps clarify the scope and implications of our study.

All things considered, the study offers insightful information about how population variance is

influenced by social status, especially multidimensional SES. It supports the emerging

knowledge that health outcomes are intricately linked to both privilege and deprivation, and

that resolving these inequities necessitates an appreciation of their complexity.

We thank the reviewers for their positive assessment of our work and thoughtful comments. We are grateful for your recognition of the study’s contribution in highlighting how multidimensional social status influences population variance. Your acknowledgment of the complex relationship between privilege, deprivation, and health outcomes is much appreciated. We agree that resolving these inequities requires a deep understanding of their multifaceted nature, and hope our study serves as a valuable step toward advancing this knowledge.

Reviewer #1: The paper emphasizes that social stratifications, particularly those linked to SES, should be understood as multidimensional and interconnected with broader social determinants of health. The study encourages considering the full spectrum of SES when studying population health and behavior. All data were made available and well defined in the body of the paper.

Reviewer #2:

1) the manuscript is technically sound and the data support the conclusion because the study identified the key drivers among the phenotypes flagged by the discovered components of major population stratification which tracks the interfamily differences.

2) the authors made all data and data source underlying the findings in their manuscript fully available.

3) the statistical analysis has been performed appropriately and rigorously and comparing ordinary PCA with CVAE component.

4) the manuscript was presented in an intelligible fashion and written in standard English, however unnecessary repetitions should be avoided.

The figures and tables are placed faraway from the discussions. Reduction in the paper size would make room for the avoidance of repetitions seen across the study discussions. Thereby enhancing the quality and understanding of the study.

Thank you for your constructive feedback. We appreciate your suggestion to reduce the distance between the figures and tables and the relevant discussions. We have reorganized the manuscript to ensure that figures and tables are placed closer to the corresponding discussions, which will improve the readability and flow of the paper. To avoid unnecessary repetitions, we have carefully gone through the entire manuscript.

Journal Requirements:

We have thoroughly reviewed the manuscript to ensure compliance with the provided style guidelines.

We have confirmed that no funding information is mentioned in the text.

“DB is a shareholder and advisory board member at MindState Design Labs, USA.”

The required statement has been added the manuscript and cover letter to confirm our adherence to all PLOS ONE policies on sharing data and materials.

4. We note that Figure 8 in your submission contain map/satellite images which may be copyrighted. All PLOS content is published under the Creative Commons Attribution License (CC BY 4.0), which means that the manuscript, images, and Supporting Information files will be freely available online, and any third party is permitted to access, download, copy, distribute, and use these materials in any way, even commercially, with proper attribution. For these reasons, we cannot publish previously copyrighted maps or satellite images created using proprietary data, such as Google software (Google Maps, Street View, and Earth). For more information, see our copyright guidelines: http://journals.plos.org/plosone/s/licenses-and-copyright.

a. You may seek permission from the original copyright holder of Figure 8 to publish the content specifically under the CC BY 4.0 license.

We have confirmed that the cartographic boundary file used in Figure 8, sourced from the U.S. Census Bureau (https://www.census.gov/geographies/mapping-files/time-series/geo/carto-boundary-file.2014.html), is permitted for public use. According to the U.S. Census Bureau’s citation policies (https://www.census.gov/about/policies/citation.html), proper attribution is required when using their data. We have ensured that the figure includes the appropriate citation in accordance with these guidelines.

Updated caption:

Fig 8. Unique SES signatures are distinctly represented in specific US states

Participant SES variable scores (n=202) in each of the 4 SES-centric components were used as features to train a multi-class logistic regression classifier to predict participant state of residence (cf. Methods). State color indicates which SES component had the strongest influence (i.e., coefficient magnitude) on predicting state of residence and shading intensity of state quantifies the magnitude of this influence. Note that component A was not most influential for any state. Different states appear to be more uniquely aligned with different SES signatures. Density plots per state show the distribution of participant scores in each of the 4 components for that state. The degree of divergence between participant score distributions vary depending on the states. The classifier was able to predict US state of residence for 10 of 17 states at above chance level (17 classes therefore chance level is ~5.88%). Map using U.S. Census Bureau Cartographic Boundary File [41].

Reference:

41. Bureau USC. Cartographic Boundary Files - Shapefile, “cb_2014_us_state_20m”. Censusgov. 2014.

Our reference list is correct and up to date. No papers have been retracted in the meantime.

---

## [Editor Report · Decision Letter 1]

Deep learning reveals that multidimensional social status drives population variation in 11,875 US participant cohort

PONE-D-24-49641R1

Dear Dr. Marotta,

We’re pleased to inform you that your manuscript has been judged scientifically suitable for publication and will be formally accepted for publication once it meets all outstanding technical requirements.

Kind regards,

Forgive Avorgbedor

Academic Editor

PLOS ONE
---

## [Editor Report · Acceptance letter]

PONE-D-24-49641R1

PLOS ONE

Dear Dr. Marotta,

I'm pleased to inform you that your manuscript has been deemed suitable for publication in PLOS ONE. Congratulations! Your manuscript is now being handed over to our production team.

Kind regards,

on behalf of

Dr. Forgive Avorgbedor

Academic Editor

PLOS ONE